# Ultrafast endocytosis at *Caenorhabditis elegans* neuromuscular junctions

**Shigeki Watanabe, Qiang Liu, M Wayne Davis, Gunther Hollopeter, Nikita Thomas, Nels B Jorgensen, Erik M Jorgensen***

Department of Biology, Howard Hughes Medical Institute, University of Utah, Salt Lake City, United States

**Abstract** Synaptic vesicles can be released at extremely high rates, which places an extraordinary demand on the recycling machinery. Previous ultrastructural studies of vesicle recycling were conducted in dissected preparations using an intense stimulation to maximize the probability of release. Here, a single light stimulus was applied to motor neurons in intact *Caenorhabditis elegans* nematodes expressing channelrhodopsin, and the animals rapidly frozen. We found that docked vesicles fuse along a broad active zone in response to a single stimulus, and are replenished with a time constant of about 2 s. Endocytosis occurs within 50 ms adjacent to the dense projection and after 1 s adjacent to adherens junctions. These studies suggest that synaptic vesicle endocytosis may occur on a millisecond time scale following a single physiological stimulus in the intact nervous system and is unlikely to conform to current models of endocytosis.

## Introduction

Synaptic transmission is quantal—postsynaptic responses have a uniform minimal size (*Fatt and Katz, 1952*). Katz hypothesized that fusion of synaptic vesicles with the plasma membrane accounted for the release of fixed amounts of neurotransmitter (*Del Castillo and Katz, 1956*). To test the 'vesicle hypothesis', two groups of scientists stimulated frog motor neurons and examined the ultrastructure by electron microscopy. After prolonged stimulation, it was noted that synaptic vesicles were depleted from the terminals, suggesting that vesicles were being consumed during synaptic transmission (*Ceccarelli et al., 1972*, *1973*; *Heuser and Reese, 1973*). Later it was found by Heuser and Reese that if the synapses were rapidly frozen after single stimulus that vesicles could be observed fusing to the plasma membrane (*Heuser and Reese, 1981*). Furthermore, Torri-Tarelli et al. demonstrated that the timing of vesicle fusion coincided precisely with that of transmitter release (*Torri-Tarelli et al., 1985*). Together, these data provided proof of the vesicle hypothesis.

These ultrastructural studies further suggested that synaptic vesicles are recycled at synapses. Recycling was originally inferred from standard fixations based on the presence of coated vesicles on the plasma membrane (*Heuser and Reese, 1973*) and the uptake of peroxidase after stimulation (*Ceccarelli et al., 1972*). Heuser and Reese improved the temporal resolution of their analysis by building a 'freeze slammer' capable of instantaneously freezing a sample at arbitrary time points after stimulation (*Heuser et al., 1979*). Invaginations were most abundant 30 s after stimulation and were observed in membrane domains lateral to the active zone in freeze fracture studies (*Miller and Heuser, 1984*). Based on the presence of coated pits in traditional fixations (*Heuser and Reese, 1973*), it was proposed that recycling of synaptic vesicles was mediated via clathrin coats.

Since the publication of the morphological data, further biochemical and functional experiments lend support for a clathrin-mediated mechanism for vesicle recycling (*Dittman and Ryan, 2009*). Reconstitution experiments demonstrate that clathrin and its adaptors are sufficient to stimulate budding from brain-derived membranes (*Takei et al., 1998*). Purification of vesicles from brain

*For correspondence:
jorgensen@biology.utah.edu

**Competing interests:** The authors declare that no competing interests exist.

**Reviewing editor**: Eve Marder, Brandeis University, United States

**eLife digest** Neurons communicate with one another at junctions called synapses. When an electrical signal travels along a neuron and arrives at a synapse, vesicles filled with small neurotransmitter molecules fuse with the cell membrane and release the neurotransmitter. These chemicals rapidly bind to receptors on the downstream neuron that induce an electrical response in that cell.

Vesicles can be consumed at prodigious rates, up to 500 a second, so the cell must recover the membrane rapidly and regenerate more vesicles filled with neurotransmitter. Experiments in the 1970s and 1980s suggested that when vesicles empty their contents into the synapse, they fuse completely with the membrane and are lost. To recover the membrane, the cell forms 'pits', by means of a coat protein called clathrin, which then bud off into the cell as new vesicles. It takes roughly 15–20 s for vesicles to be recycled in this way. By contrast, synapses with very high firing rates are thought to recycle vesicles through a faster process known as 'kiss and run', in which vesicles are not fully integrated into the membrane, but instead fuse transiently with it to form a reversible pore within about a second.

However, these studies triggered vesicle release using conditions that are unlikely to occur naturally inside cells. Now, Watanabe et al. have used optogenetics to study vesicle recycling in response to single stimuli at the synapse between neurons and muscles in an intact living animal, the nematode *C. elegans*. The worms had been genetically modified to express a light-sensitive ion channel called channelrhodopsin in their motor neurons. Watanabe et al. used a single pulse of light to stimulate vesicle release, and then rapidly froze the worms before studying their synapses with electron microscopy.

They found that vesicle recycling occurred at the edges of the synapse or at a specialized structure in the middle of the synapse. Vesicle recycling took less than 50 ms—much faster than anything previously observed. This ultrafast recycling is unlikely to occur via 'kiss and run' since recycling occurred at sites lateral to the sites of fusion and because the recycled vesicles were larger than the originals, implying that they had not simply re-formed after a brief fusion event.

By using physiologically relevant stimuli in an intact animal, Watanabe et al. reveal that vesicles can be recycled at synapses much more rapidly than previously thought, suggesting that our current models of this process may need to be reassessed.

demonstrated that synaptic vesicle proteins (and not just membrane) are found in clathrin-coated vesicles (*Maycox et al., 1992*). Furthermore, clathrin-coated pits accumulate at terminals in which vesicle scission is blocked by interfering with dynamin (*Koenig and Ikeda, 1989*; *Shupliakov et al., 1997*). Disruption of clathrin function by peptide injections, siRNA application, or photoinactivation all slow the rate of endocytosis (*Jockusch et al., 2005*; *Granseth et al., 2006*; *Heerssen et al., 2008*; *Kasprowicz et al., 2008*). Finally, synaptic vesicle proteins tagged with pHlourin are recycled in about 15 s, roughly the same speed as membrane recycling observed in ultrastructural experiments (*Granseth et al., 2006*; *Balaji and Ryan, 2007*).

Clathrin-mediated endocytosis, however, may not constitute the sole pathway for synaptic vesicle endocytosis. The prodigious rate of exocytosis at some synapses may exceed the speed of clathrin-mediated endocytosis. For example, during locomotion, lumbar motor neurons in the mouse can fire at ~100 Hz (*Maeno-Hikichi et al., 2011*). Moreover, in some central synapses such as the Calyx of Held, firing rates can reach up to 500 Hz (*Kopp-Scheinpflug et al., 2008*; *Lorteije et al., 2009*). Calculations suggest that vesicles are reused every 13 s at the Calyx of Held, assuming that all the vesicles in the terminal are mobile (*Neher, 2010*). This rate of reuse is faster than the 15–20 s required to recover vesicles by clathrin-mediated endocytosis at synapses (*Miller and Heuser, 1984*; *Cocucci et al., 2012*), and would overwhelm the capacity to recycle vesicles. Furthermore, experiments using FM dyes suggest that only 5–20% of vesicles in the terminal are cycling during physiological activity (*Harata et al., 2001*; *Rizzoli and Betz, 2005*; *Denker et al., 2011*; *Marra et al., 2012*). Therefore, even fewer vesicles may be available to sustain high release rates, placing an even greater demand on the endocytic machinery (*De Lange et al., 2003*; *Neale et al., 1999*).

An alternative mechanism for synaptic vesicle endocytosis, now called 'kiss-and-run' (*Fesce et al., 1994*), was proposed by Bruno Ceccarelli based on experiments nearly identical to those of Heuser and Reese. When frog neuromuscular junctions were stimulated at low frequency, vesicles with fusion pores lacking clathrin coats were observed in the active zone (*Ceccarelli et al., 1972*). The fusion pores are unstable, lasting about a second (*Wu and Wu, 2009*; *Zhang et al., 2009*). Some experiments using optical tracers including FM dyes, pH-sensitive fluorescent proteins, and quantum dots support a fast pathway (*Richards et al., 2000, 2005*; *Aravanis et al., 2003*; *Harata et al., 2006*; *Zhang et al., 2007, 2009*; *Zhu et al., 2009*; *Park et al., 2012*). This has led to a model in which exocytosis and endocytosis are coupled via a transient fusion pore. The key aspects of kiss-and-run endocytosis is that first, the vesicles are recycled at the site of release in the active zone; second, they are recovered quickly (in less than a second), and third, the vesicles remain intact and retain their protein and membrane complement, rather than fully collapsing into the membrane.

Under high-frequency stimulation bulk endocytosis contributes to vesicle recycling (*Smith et al., 2008*; *Cousin, 2009*). During repetitive stimulation a large amount of membrane is added to the surface, which creates deep invaginations into the cell. These structures are pinched off by dynamin, and synaptic vesicles are then regenerated from these endosomes (*Miller and Heuser, 1984*; *Cousin, 2009*). Bulk endocytosis has been observed following intense stimulation at frog neuromuscular junctions (*Miller and Heuser, 1984*; *Richards et al., 2000*; *Gaffield et al., 2011*), retinal bipolar cells (*Holt et al., 2003*), and mammalian central synapses (*Clayton et al., 2010*).

Ultrastructural analysis of stimulated neurons has provided the basis for models of synaptic vesicle endocytosis. However, in each of these studies, non-physiological or intense sustained stimulation was used in dissected preparations to trigger cycling of synaptic vesicles. For example, for cross sections of the frog neuromuscular junction, Heuser and Reese applied stimulation at 10 Hz for 1 min (1973), and Ceccarelli employed 2 Hz stimulation for 2 hr (1972). For freeze-fracture studies, Heuser and Miller delivered a single stimulus in the presence of 4-aminopyridine and 10 mM calcium; these images provide the best temporal analysis of endocytosis but the nature of the invaginations was not examined in cross sections. Here, we stimulated motor neurons in vivo with a single activation pulse by using the light-sensitive ion channel ChIEF (*Lin et al., 2009*). Samples were rapidly frozen using a high-pressure freezer, and the ultrastructure was examined using transmission electron microscopy. We found that endocytosis occurs rapidly at two sites, adjacent to the dense projection at the center of the synapse and at adherens junctions flanking the synapse. These two regions represent the edges of the active zone, and endocytosis is complete within 50 ms and 3 s, respectively, after stimulation.

## Results

### Stimulation of neurons with millisecond temporal resolution

Capturing membrane dynamics in an intact organism requires rapid freezing of tissue. Heuser and Reese used a freeze-slammer to capture membrane dynamics after synchronous neurotransmitter release (1979 and 1981). However, freeze-slamming cannot preserve morphology deeper than 10 μm from the cell surface without ice crystal damage, whereas the diameter of an adult worm is about 70 μm. We employed high-pressure freezing because it can freeze a sample as thick as 200 μm within 25 ms with reduced ice crystal formation (*Moor and Riehle, 1968*; *Moor, 1987*).

The second requirement for an ultrastructural analysis of vesicle dynamics is that the synapse must be stimulated at a precise time relative to freezing. Previously, we demonstrated that channelrhodopsin-2 (*Nagel et al., 2003*) can be expressed in *Caenorhabditis elegans* motor neurons to stimulate neurotransmission with pulses of light (*Liu et al., 2009*). Here, we expressed ChIEF, a variant of channelrhodopsin that is less prone to desensitization (*Lin et al., 2009*), in acetylcholine motor neuron. We applied a single pulse of blue light to animals while recording currents from the post-synaptic muscle cell (*Figure 1A*). The amplitudes of evoked responses triggered by ChIEF are similar to electrically stimulated responses, suggesting that ChIEF activation can cause the simultaneous fusion of about 100 synaptic vesicles per muscle cell or approximately 4–10 synaptic vesicles per synapse.

Finally, we created a light path to the specimen by drilling a hole through the bayonet specimen holder for the Leica EMpact2 high-pressure freezer; we replaced the black diamond anvil with a transparent sapphire anvil (*Figure 1B, C*). We mounted a 3 mm LED at the tip of the bayonet (*Figure 1C*); the measured light intensity at the sample was ~20 mW/mm². We constructed a programmable circuit with a temporal resolution of ~5 ms or less. The minimal interval between stimulation and freezing was

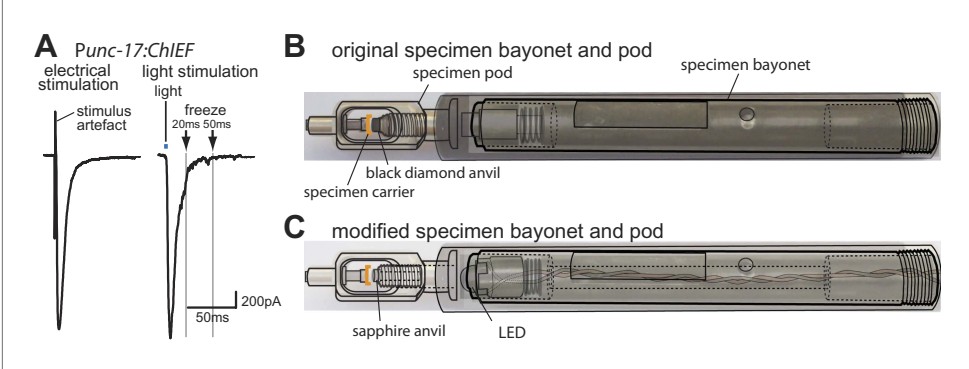

**Figure 1**. Optogenetics coupled with high-pressure freezing. (**A**) A sample trace of an excitatory postsynaptic current from the muscle of animals expressing ChIEF in acetylcholine neurons, evoked by a single depolarizing pulse (left) and by a 3 ms pulse of blue light (right). The blue line indicates the application of blue light. Arrows indicate freezing times at 20 ms and 50 ms. (**B**) A schematic diagram of original specimen bayonet and a pod. Specimens in the cup of the specimen carrier face the black diamond anvil. (**C**) A schematic diagram of modified specimen bayonet and a pod. The bayonet was bored out (the dotted lines) to house the LED and wires. A sapphire end stone was mounted into the pod. Specimens in the cup of the specimen carrier face the sapphire anvil and LED.

~20 ms due to the mechanics of the freezing device ('Materials and methods'). The motor neurons were depolarized by a light pulse and rapidly frozen under high pressure at fixed time points. At each time point serial sections from 12–26 acetylcholine synapses were imaged using an electron microscope, and morphometry of axon profiles was conducted blind to genotype.

## Only docked vesicles are released by depolarization

The site of vesicle fusion is defined as the active zone (***Couteaux and Pécot-Dechavassine, 1974***; ***Landis, 1988***). To establish the relationship between sites for vesicle fusion and for recycling, we determined the extent of the active zone at neuromuscular junctions in *C. elegans* (***Figure 2***). Vesicles are morphologically docked along a broad stretch of the plasma membrane juxtaposed to the muscle and are considered to comprise the readily releasable pool (***Hammarlund et al., 2007***). We froze animals 20 ms after stimulation and observed vesicles fusing along the plasma membrane (***Figure 2A–E***); the diameter of these vesicles was 28.4 nm ± 0.3, the same size as a synaptic vesicle. These exocytic pits were not observed in the non-stimulated control and at 30 ms after stimulation after postsynaptic currents have ended (***Figure 1A***). The exocytic intermediates have wide openings at their necks, suggesting that vesicles fully collapse into the membrane. The number of docked vesicles was reduced after the stimulus (***Figure 2G,H***), whereas the number of tethered vesicles remained the same (***Figure 2I,J***). These data suggest that docked, but not tethered, vesicles fuse in response to depolarization and thus constitute the readily releasable pool (***Zenisek et al., 2000***).

## Vesicles fuse across a broad surface of the synapse during evoked release

Fusing vesicles were found along the entire face of the synapse between the dense projection and the flanking cell junctions; thus the active zone for the *C. elegans* neuromuscular junction is rather broad (***Figure 2A–F*** and ***Figure 2—figure supplement 1***). We quantified these events per synaptic profile, that is, in cross-sections of a synapse that contained a portion of the dense projection. On average, approximately one fusion intermediate was observed per synaptic profile although this number could be an underestimate given that neurotransmission is declining at this time point (***Figure 1A***).

To estimate the total number of vesicle fusions per synapse, we froze animals 50 ms after stimulation when neurotransmission is complete (***Figures 1A and 3***). Fusion intermediates were no longer observed 50 ms after stimulation; however, docked vesicles were depleted from the terminals (***Figure 3***; more micrographs in ***Figure 3—figure supplement 1A–C***). These results suggest that fusing vesicles had collapsed into the membrane by 50 ms but that release sites had not been refilled. We estimated the total number of vesicle fusions and determined the release sites by comparing docked vesicles

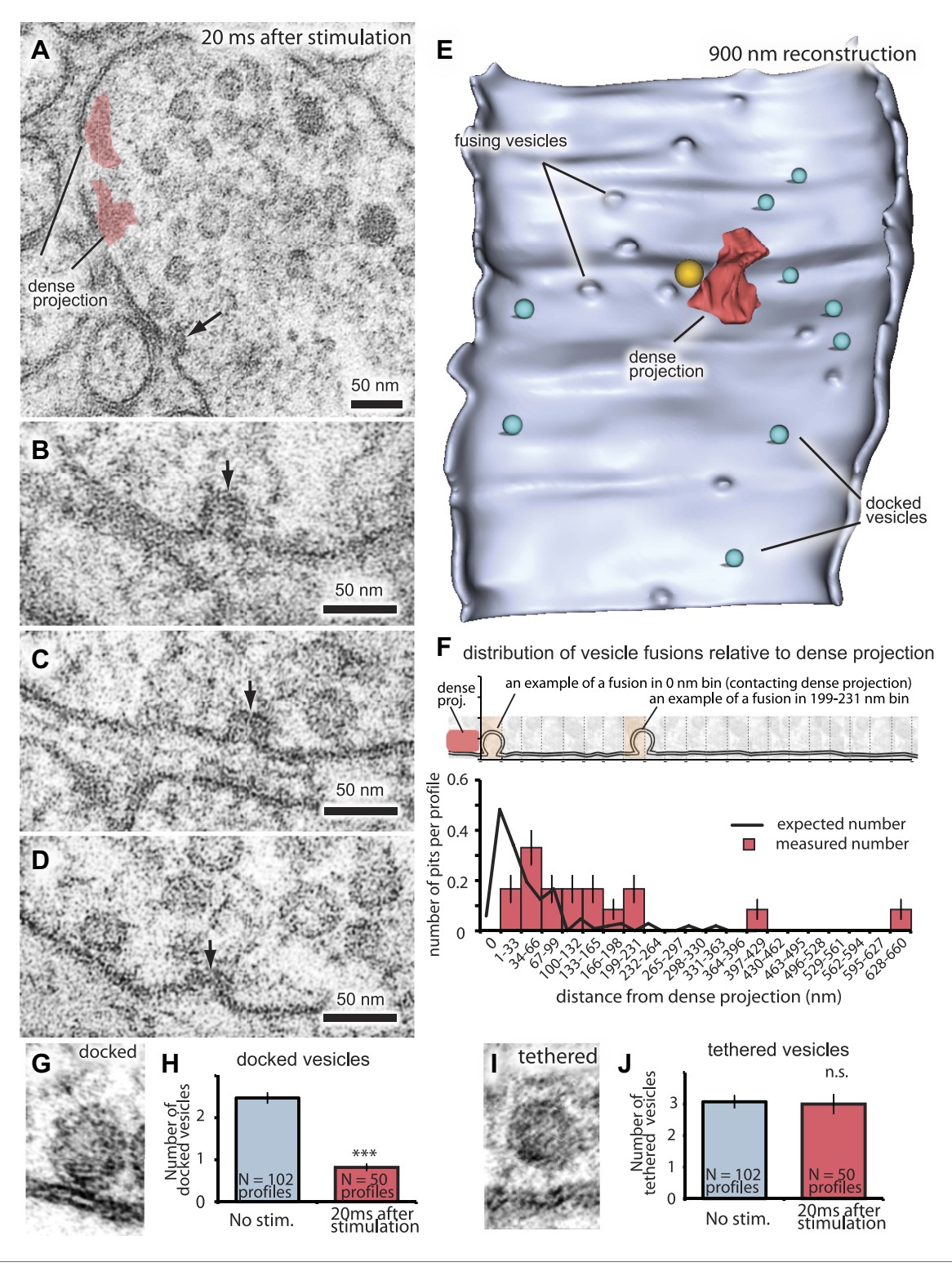

**Figure 2**. Fusing vesicles are observed 20 ms after stimulation. (**A–D**) Sample electron micrographs of acetylcholine neuromuscular junctions 20 ms after depolarization by light stimulation of channelrhodopsin. Fusing vesicles, indicated by the black arrow, with the size of a synaptic vesicle were present in the active zone. (**B–D**) The necks of these pits are wide, suggesting that vesicles are undergoing full collapse into the membrane. (**E**) An active zone stimulated at 20 ms was reconstructed from 27 serial sections for a total distance of ~900 nm. The reconstruction is cut along the adherens junctions, and shows the entire face of the active zone. Docked vesicles are colored green. A large vesicle (orange) was observed above the dense projection (red) at
*Figure 2. Continued on next page*

*Figure 2. Continued*

this synapse. (**F**) The distance of fusing vesicles from the dense projection per profile. A total of 17 exocytic intermediates was scored from 50 synaptic profiles analyzed. The inset illustrates how vesicles were placed into 33 nm bins from the dense projection. The column labeled as 0 nm includes vesicles that are touching the dense projection. Orange shading shows examples of binning. The black line indicates the expected number and fusion sites of vesicles based on the loss of docked vesicles at 20 ms (**Figure 3G**). Note that fusing vesicles were not observed in proximity of the dense projection (within 33 nm) likely because they had collapsed before freezing at the 20 ms time point. Representative micrographs of (**G**) a docked vesicle (in direct contact with the plasma membrane), and (**I**) a tethered vesicle (within 30 nm of plasma membrane). (**H** and **J**) Average number of vesicles per profile docked (**H**) and tethered (**J**) along the active zone. Only the number of docked vesicles was reduced after stimulation. The number is normalized by the length of active zone. ***, **, * indicate p-values of <0.001, <0.01, and <0.05, respectively. n.s., 'not significant'. The standard error of the mean is shown in each graph.

The following source data and figure supplements are available for figure 2:

**Source data 1**. *Figure 2H,J*: The numbers of docked and tethered vesicles in each profile were normalized by the area of active zones.

**Figure supplement 1**. *C. elegans* neuromuscular junction has a broad active zone.

before and after stimulation. In unstimulated neurons, there were about 11 docked vesicles in the active zone lateral to the dense projection. A single stimulus caused the release of about six to seven vesicles from profiles containing a dense projection, or about one third of the docked vesicles in these sections. These synapse reconstructions are only partial. From full end-to-end reconstructions of synapses, we previously counted a mean of 34 docked vesicles per synapse (**Hammarlund et al., 2007**). Thus, a single neuromuscular junction may release up to 12–18 vesicles per stimulus, in agreement with our reconstructed synapse 20 ms after stimulation (**Figure 2E**).

To determine the location of release sites, we analyzed the distribution of docked vesicles along the plasma membrane relative to the presynaptic dense projection. Following a single stimulus, docked vesicles were depleted along the large surface area of the synapse that faces muscle cells (**Figure 2E** and **Figure 3C,G**). Some vesicles are docked in the perisynaptic zone beyond the adherens junctions at the far edge of the synapse, sometimes called 'ectopic docking' (**Zenisek et al., 2000**; **Lenzi et al., 2002**; **Matsui and Jahr, 2003**; **Hammarlund et al., 2007**). These perisynaptic vesicles are not released after a stimulus (**Figure 3F**). Together, these results suggest that the active zone extends on either side of the dense projection to the flanking cell-cell junctions (**Figure 3C**).

## Refilling the docked pool

If the pool of docked vesicles solely constitutes the readily-releasable pool, then the morphologically defined docked pool should be restored in parallel with the electrophysiologically defined readily-releasable pool. Acetylcholine neuromuscular junctions in *C. elegans* exhibit rapid depression when stimulated at high frequency (**Liu et al., 2009**). To determine the refilling rate of the readily releasable pool, we characterized paired-pulse depression at acetylcholine neuromuscular junctions. We applied two 10 ms light pulses to neuromuscular junctions of transgenic animals (*acr-16(ok789)*; P*unc-17::ChIEF::mCherry*) while patch-clamping a body wall muscle. Nicotine-sensitive acetylcholine receptors (encoded by *acr-16*) desensitize rapidly, and their resensitization would distort the time constant for the recovery of the readily-releasable pool. Therefore, the paired-pulse recordings were performed in animals lacking nicotine-sensitive but retaining the non-desensitizing levamisole-sensitive acetylcholine receptors. By varying the time of the application of the second pulse, the recovery rate from synaptic depression was measured (**Figure 4A**). The time constant (τ) for the slow phase of recovery was 2.2 s, and required about 6 s for full recovery (**Figure 4B**).

To determine the time course for the recovery of docking, we characterized the ultrastructure of synapses after single stimuli (**Figure 4C–F** and **Figure 3—figure supplement 1**). Acetylcholine motor neurons were depolarized with a single light pulse, allowed to recover for 20 ms, 50 ms, 100 ms, 300 ms, 1 s, 3 s, and 10 s, and then frozen (**Figure 3—figure supplement 1A–H**). The number of docked vesicles was reduced from 2.6 to 1 immediately after the stimulus (20–300 ms: **Figure 4D**). Docked vesicles were not depleted in transgenic worms without trans-retinal (the second column in **Figure 4D,E**), suggesting that the depletion of vesicles depended on the activation of channelrhodopsin. Docked vesicles in active zones recovered with a time constant of 2.4 s (**Figure 4C**), which corresponds well to the rate of electrophysiological recovery. The total number of vesicles, as well as tethered vesicles,

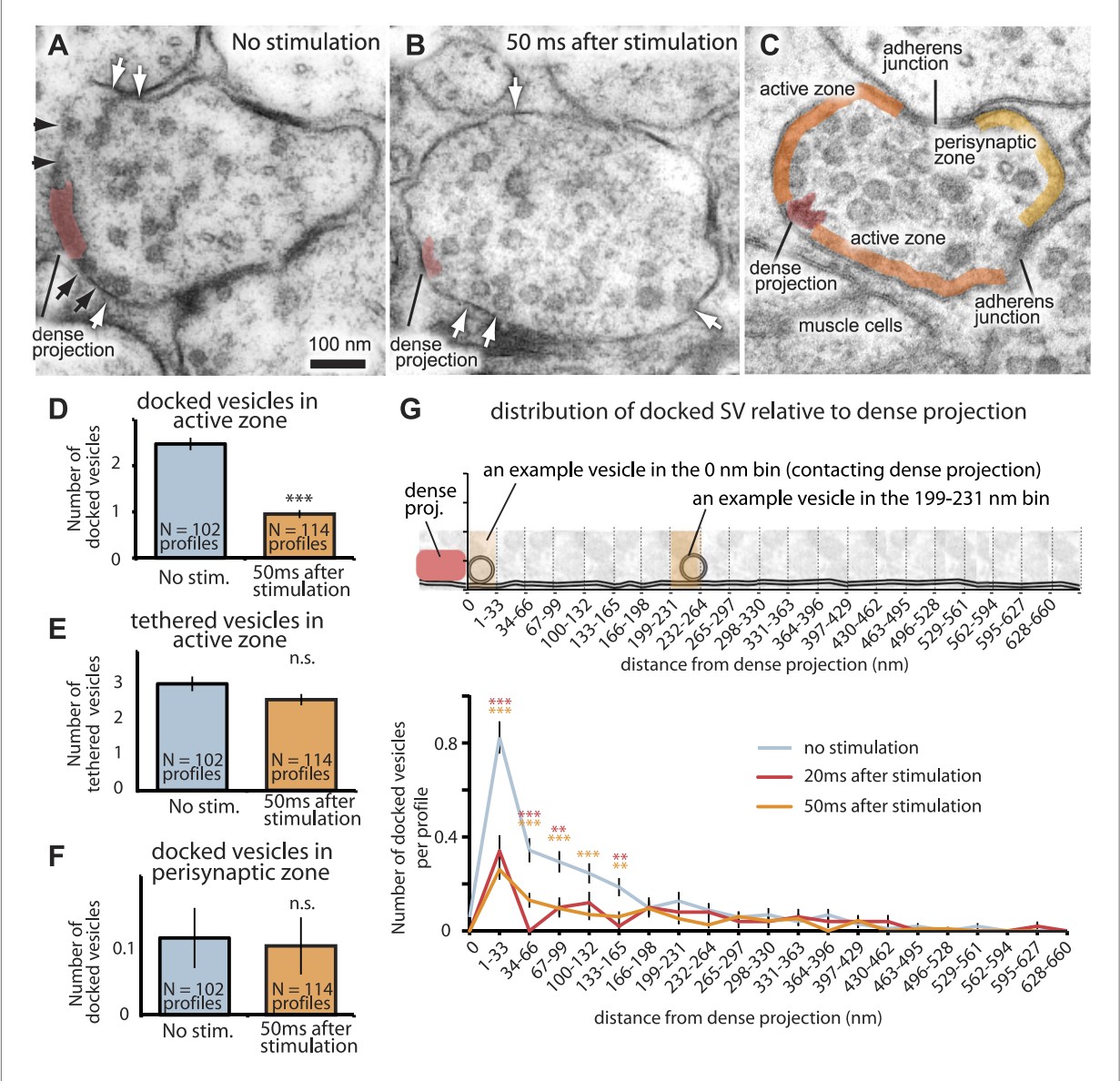

**Figure 3**. The readily-releasable pool is solely constituted of docked vesicles. An electron micrograph of an unstimulated (**A**) and stimulated (**B**) acetylcholine motor neuron after depolarization by channelrhodopsin. Docked vesicles (black arrows) are present near the dense projection before the stimulation but are absent 50 ms after the stimulation. Tethered vesicles (white arrows) are present before and after stimulation. (**C**) The active zone extends from the dense projection to the flanking adherens junctions to neighboring axons at *C. elegans* neuromuscular junctions based on fusion sites. Note that the side of the neuron that faces the muscle (bottom) can be very broad. (**D**) Average number of vesicles docked in active zones per profile. A significant reduction in docked vesicles was observed after depolarization (unstimulated 2.5± 0.1 SV per profile; stimulated 0.9 ± 0.1 SV; p<0.0001). The number is normalized by the length of the active zone. (**E**) Average number of vesicles per profile tethered in the active zone. No reduction in tethered vesicles was observed after depolarization (unstimulated 3.1 ± 0.2 SV per profile; stimulated 2.6 ± 0.2 SV; p=0.21). The number is normalized by the length of the active zone. (**F**) Average number of docked vesicles per profile in the perisynaptic zone before and after stimulation, normalized by the length of the perisynaptic zone. The docked vesicles in the perisynaptic zone were not released by a stimulus (p=0.95). (**G**) Distance of docked vesicles to the dense projection in the active zone. The absolute number of docked vesicles per synaptic profile is shown. Docked vesicles are released across the entire length of the active zone. The binning of vesicle distances is similar to *Figure 2F*. The asterisks indicate a significant difference in the number of docked vesicles between stimulated and unstimulated at a particular distance from the dense projection. Note that the adherens junctions are located on average 250–350 nm from the dense projection, but in acetylcholine neurons the arrangement is asymmetric–the active zone facing the muscle can be up to ~1000 nm. Thus, the apparent reduction in docked vesicles further than 133–660 nm is caused by the staggered ends of the active zones rather than a reduction in the density of docked vesicles. Docked vesicles in the perisynaptic zone are not counted. ***, **, * indicate p-values of <0.001, <0.01, and <0.05, respectively. 'n.s.', not significant. The standard error of the mean is shown in all graphs.

*Figure 3. Continued on next page*

*Figure 3. Continued*

The following source data and figure supplements are available for figure 3:

**Source data 1**. *Figure 3D,E,F*: The numbers of docked and tethered vesicles in each profile were normalized by the area of active zones or perisynaptic zone.

**Figure supplement 1**. The readily-releasable pool is solely constituted of docked vesicles.

decreased slightly after stimulation but recovered by 300 ms (*Figure 4E,F*) and can be accounted for by the number of vesicles lost through fusion. This fluctuation may indicate that vesicles pass through a tethered intermediate (*Zenisek et al., 2000*). Together, these results suggest that the docked vesicles are likely the morphological correlate of the readily releasable pool.

## Invaginations adjacent to the dense projection 20 ms after stimulation

Following exocytosis, synaptic vesicle proteins and membranes need to be recycled locally at synapses. In the kiss-and-run model vesicles do not collapse into the membrane, but instead the fusion pore reverses rapidly (within 1 s; *Zhang et al., 2009*) to regenerate the vesicle. Clathrin-mediated endocytosis is thought to take place lateral to the active zone, by the relatively slow (~20 s; *Miller and Heuser, 1984*) recruitment of clathrin machinery (*Rizzoli and Jahn, 2007*; *Dittman and Ryan, 2009*). Images from three different animals frozen at each time point were scored blind for endocytic structures (*Figure 5*). In these experiments, serial sections that contained a cross-section of the dense projection were reconstructed, usually 4–6 synaptic profiles, and together are defined as a 'synapse' for these experiments. At 20 ms after stimulation, about 50% of the synapses exhibited large invaginations next to the dense projection (*Figure 5B,D,G and H*, and *Figure 5—figure supplement 1*). These invaginations were present 30 ms after stimulation but were completely absent by 50 ms. Large vesicles were adjacent to the dense projection in 50% of the synapses at 20 ms, their frequency increases at 30 ms, and peaks at 50 ms (*Figure 5B*). At least one large vesicle but usually more were found adjacent to each dense projection at 50 ms after stimulation (*Figure 5B,E and F*; more examples in *Figure 5—figure supplement 1*). These large vesicles appeared only after stimulation (*Figure 5B*), suggesting that the formation of large vesicles is triggered by exocytosis. The average diameter of the large invaginations and vesicles near dense projection (42.8 nm) was significantly larger than synaptic vesicles (29.3 nm; p<0.0001) or fusing vesicles (28.4 nm; p<0.0001; *Figure 5A*) and the diameter was bimodally distributed (*Figure 5I*). Large vesicles at the dense projection declined after 50 ms, and large vesicles peaked in the center of the synaptic varicosity between 100 ms and 300 ms. The large vesicles in the center of the synapse declined to background levels by 3 s suggesting that they are resolved into smaller vesicles, potentially synaptic vesicles.

To test if endocytosis at the dense projection is triggered by exocytosis and not by calcium influx alone (*Neale et al., 1999*), we stimulated synapses in *unc-13(s69)* mutants (*Figure 5—figure supplement 2*). Exocytosis is almost completely absent in *unc-13* mutants (*Richmond et al., 1999*), presumably due to the lack of docked vesicles (*Hammarlund et al., 2007*). Therefore, the stimulation of the neuron in an *unc-13* mutant should cause an influx of calcium without the addition of membrane to the surface. There was no increase in the number of large vesicles next to the dense projection after stimulation (*Figure 5C*), although occasionally a large vesicle was observed in unstimulated *unc-13* mutants. Thus, the formation of large vesicles at the dense projection requires exocytosis of synaptic vesicles.

## Membrane invaginations at adherens junctions 300 ms after stimulation

After about 100 ms, invaginations appeared at the flanking adherens junctions (*Figure 6A,E, and G*; more example micrographs in *Figure 6—figure supplement 1A–C*). At later time points, deeper invaginations and large vesicles (*Figure 6B,C*; more example micrographs in *Figure 6—figure supplement 1D–F*) were observed at adherens junctions. These invaginations appeared to lack clathrin coats, although clathrin-coated vesicles can be observed using this sample preparation (*Figure 6D* and *Figure 6—figure supplement 1H*). The mean diameter of deep pits, as well as large vesicles, was 44 nm, the equivalent of 2.3 synaptic vesicles (*Figure 6—figure supplement 1G*). The prevalence of pits peaks at 300 ms (*Figure 6E*) and decays with a time constant of 1.4 s (*Figure 6F*). Large vesicles associated with the adherens junction (within 50 nm) peak at 3 s and decline, suggesting that pits are

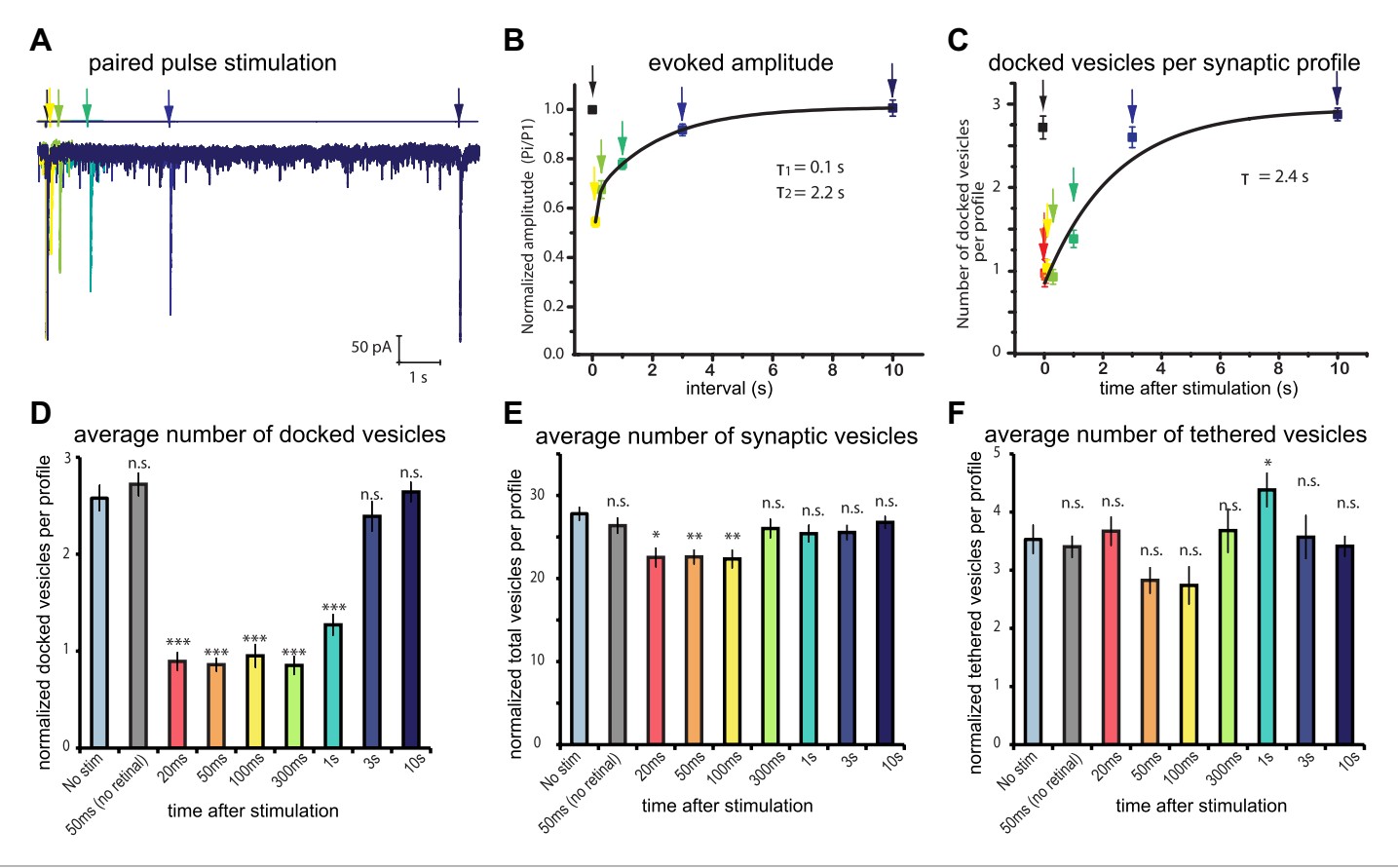

Figure 4. Physiological recovery corresponds to refilling of docking sites. (A) Sample traces for paired-pulse stimulations using channelrhodopsin recorded from an adult body muscle (*acr-16(ok789); oxSi91[Punc17::ChIEF::mCherry]*). The second pulses are indicated by colors: yellow, 100 ms; green, 300 ms; emerald, 1000 ms; blue, 3000 ms; and dark blue, 10000 ms. (B) Recovery of evoked responses in paired pulse stimulations. The ratios of the second current peak ($P_i$) to the first current peak ($P_1$) were calculated and plotted relative to the recovery intervals. There are two time constants for recovery: the first tau was 0.1 s, and the second tau was 2.2 s. The rapid recovery ($\tau$ = 100 ms) is likely due to a transition of docked vesicles from an incompetent to competent state (*Zenisek et al., 2000*), rather than the translocation and docking of vesicles. (C) Average number of vesicles per profile docked in the active zone is plotted relative to the intervals between stimulation and freezing. The time constant for the recovery was 2.4 s. (D) Average number of vesicles docked per profile at various time points after the stimulus. Docked vesicles were reduced at time points before 3 s. (E) Total number of synaptic vesicles, averaged per profile. A slight reduction in the total number of synaptic vesicles was observed after stimulation. (F) Number of vesicles per profile tethered in the active and perisynaptic zones. The size of the tethered pool drops initially after stimulation. The numbers in (D) and (F) include vesicles in both the active and perisynaptic zones. The numbers in (D), (E) and (F) are normalized by the area of the profiles to account for differences in the size of varicosities ('Materials and methods'). All the p values were calculated against no stimulation control. Bonferroni correction was applied for multiple comparisons. For detailed numbers and statistical analysis, *Figure 4—source data 1*. ***, ** and * indicate p-values of <0.0001, <0.001 and <0.007, respectively. n.s., 'not significant'. N values: not-stimulated: 102 profiles; no retinal 50 ms: 51 profiles; 20 ms: 50 profiles; 50 ms: 104 profiles; 100 ms: 83 profiles; 300 ms: 89 profiles; 1 s: 111 profiles; 3 s: 91 profiles; 10 s: 121 profiles. The standard error of the mean is shown in each graph.

The following source data are available for figure 4:

Source data 1. *Figure 4D,E,F*: The numbers of docked and tethered vesicles in each profile were normalized by the area of profiles.

precursors to large vesicles. After 10 s, the number of large vesicles adjacent to the adherens junction declines to background levels, and vesicles in the center of the terminal peak at this time (dashed line in *Figure 6E*), suggesting that the vesicles were transported to the center of the terminal. Endocytic pits did not appear in animals lacking trans-retinal or in *unc-13(s69)* mutants (*Figure 6E* and *Figure 5—figure supplement 2*), thus the formation of pits was triggered by the exocytosis of synaptic vesicles. Together, these results suggest that a second endocytosis pathway takes place at adherens junctions.

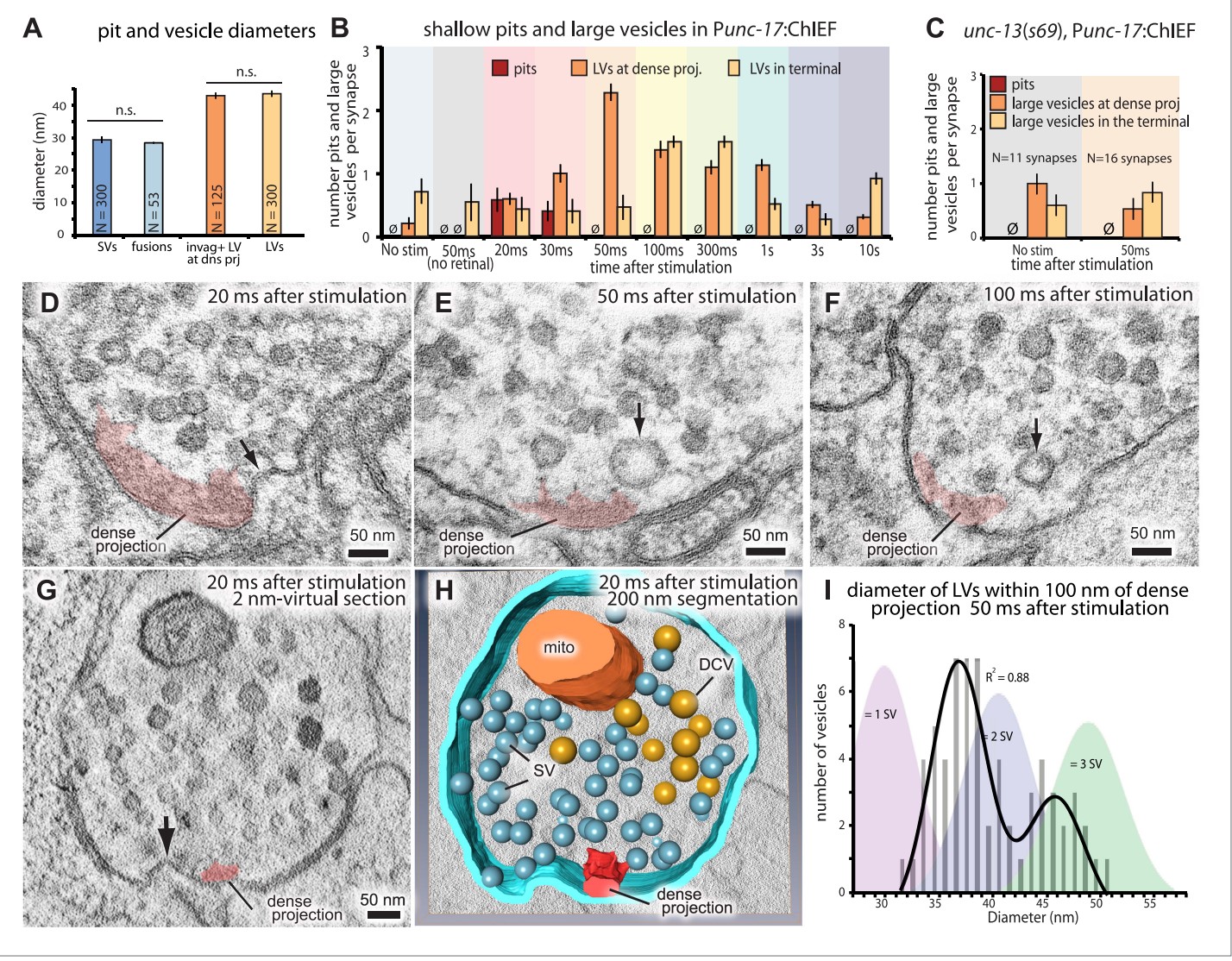

**Figure 5**. Ultrafast endocytosis takes place near the dense projection. (**A**) Average size of synaptic vesicles, exocytic pits in active zone, large invaginations at dense projections, and large vesicles in the terminal. The diameter of invaginations and large vesicles (LV) was ~1.5 times larger than that of exocytic pits observed at 20 ms after stimulation. (**B**) Time course of endocytosis at the dense projection. Structures are quantified for individual dense projections (a 'synapse') reconstructed from serial electron micrographs. Large vesicles are defined as clear-core vesicles with a diameter larger than 35 nm. Large vesicles at the dense projection are within a radial distance of 100 nm from the dense projection. (See ***Figure 5—source data 1*** for numbers and statistics.) (**C**) Pits and large vesicles at the dense projection in *unc-13 (unc-13(s69)*; P*unc-17::ChIEF::mCherry*) animals in unstimulated animals or animals frozen 50 ms after stimulation. Large vesicles are occasionally found in the synapses of unstimulated animals. (**D–F**) Electron micrographs showing an invagination at the dense projection of an acetylcholine motor neuron 20 ms after stimulation (**D**) and large vesicles near dense projections 50 ms (**E**) and 100 ms (**F**) after stimulation. (**G–H**) An electron tomogram and a hand-segmented model of an acetylcholine motor neuron synapse captured 20 ms after stimulation. An invagination of membrane is visible next to the dense projection. (**I**) The numbers of large vesicles within 100 nm of a dense projection are plotted against their diameter. The background shading indicates the expected diameters for vesicles composed of 1, 2 or 3 synaptic vesicles. There are two populations of large vesicles: one peaks at 38 nm and the other at 46 nm. $R^2$ value for bimodal distribution was 0.88 whereas that for unimodal was 0.78. Akaike information criterion for unimodal, bimodal, and trimodal distributions were 352.6, 346.4, and 353.9, respectively, supporting the bimodal distribution. Interestingly, the difference between the peaks is equivalent to the amount of membrane in a single synaptic vesicle. N for each condition in (**A**) and (**B**) was as follows: non-stimulated: 26 synapses; no retinal 50 ms: 12 synapses; 20 ms: 12 synapses; 30 ms: 12 synapses; 50 ms: 24 synapses; 100 ms: 19 synapses; 300 ms: 20 synapses; 1 s: 23 synapses; 3 s: 19 synapses; 10 s: 26 synapses; *unc-13* unstimulated: 11 synapses; *unc-13* stimulated: 14 synapses. The standard error of the mean is shown in each graph.

*Figure 5. Continued on next page*

*Figure 5. Continued*

The following source data and figure supplements are available for figure 5:

**Source data 1**. *Figure 5A*: Average size of synaptic vesicles, exocytic pits in active zone, large invaginations at dense projections, and large vesicles in the terminal. *Figure 5B*: Number of endocytic structures.
**Figure supplement 1**. Ultrafast endocytosis takes place near the dense projection.
**Figure supplement 2**. Endocytosis is coupled to exocytosis.

## Dynamin is required for fast endocytosis

Endocytic structures are often cleaved from the surface by dynamin (*Koenig and Ikeda, 1989*; *Shupliakov et al., 1997*). To test if dynamin is also required to resolve ultrafast endocytic intermediates at these sites into vesicles, *dyn-1(ky51)* animals were stimulated with a single light pulse and then frozen 1 s later. *dyn-1(ky51)* is a temperature-sensitive allele which is not functional at 30°C (*Clark et al., 1997*). Following the exposure to non-permissive temperature (30°C) for 2 min, large vesicles were observed adjacent to the dense projection and adherens junctions in the non-stimulated control (*Figure 7A–C*; more example micrographs *Figure 7—figure supplement 1*). These vesicles were often associated with a filament to the dense projection or membrane. Although necks were difficult to discern, prolonged incubation at the restrictive temperature (5 min) revealed the presence of membrane necks attached to the plasma membrane (*Figure 7—figure supplement 1C*). Typically, 2.7 ± 0.4 large vesicles were observed in each synapse (*Figure 7D*). The stimulation in these animals did not increase the number of large vesicles (2.7 ± 0.6; *Figure 7D*). However, the diameter of these structures became 1.5 times larger (from 59.0 ± 5.9 nm to 85.7 ± 8.5 nm; *Figure 7E–G*). The amount of surface area incorporated into these large vesicles equals to ~12 synaptic vesicles—the number of synaptic vesicles estimated to fuse with a single stimulus (*Figure 2E*, 'Materials and methods'). These data suggest that dynamin is required to cleave the structures from the surface and that the dense projection and the adherens junctions are the endocytic sites in *C. elegans* neuromuscular junctions.

## Membrane retrieval is complete within 10 s after a single stimulus

Does fast endocytosis account for the number of vesicles that fuse during neurotransmission? We calculated the total amount of membrane exocytosed and endocytosed at 300 ms and at 1 s after stimulation in our profiles ('Materials and methods'). We estimate that a total of 132–154 synaptic vesicles fused to the plasma membrane for all 22 synapses analyzed at the 300 ms time point and a total of 111 synaptic vesicle equivalents were recycled. For the 1 s time point, a total of 138–161 synaptic vesicles were estimated to exocytose, and membrane area equivalent to roughly 144 synaptic vesicles was recovered. These data suggest that most vesicles that fuse during exocytosis are retrieved rapidly after exocytosis; 43% are recovered at dense projections with a time constant of less than 50 ms and 57% are recovered at adherens junctions with a time constant for pit resolution of 1.4 s (*Figure 6F*).

## Discussion

Two major models for synaptic vesicle endocytosis have emerged from studies of ultrastructure: clathrin-mediated endocytosis and kiss-and-run endocytosis. Clathrin-mediated endocytosis predicts that vesicles collapse into the membrane and are recovered slowly at a specialized site lateral to the active zone. Kiss-and-run predicts that vesicles fuse briefly to the membrane and are recovered at the site of release. A long-standing criticism of these morphological studies has been that they did not employ single physiological stimuli, but rather relied on either high-frequency stimulation or pharmacological methods to increase exocytosis. A brief review is warranted: *Heuser and Reese (1973)* stimulated neuromuscular junctions at 10 Hz for 1 min and then fixed the sample using ice-cold glutaraldehyde. After this intense stimulation, vesicles were depleted and invaginations with distinct clathrin coats appeared on the plasma membrane. *Miller and Heuser (1984)* stimulated the nerve with a single stimulus in the presence of the $K^+$ channel blocker 4-aminopyridine and 10 mM calcium and froze the sample rapidly using a freeze-slammer. 20 s after this single stimulus, pits formed at the lateral edges of the active zone. Because these were freeze-fracture studies, the presence of clathrin coats could not be ascertained. However, because clathrin coats had been observed in the 1973 study, they concluded that synaptic vesicle endocytosis is most likely mediated by clathrin (*Heuser and Reese, 1979*; *Miller and Heuser, 1984*).

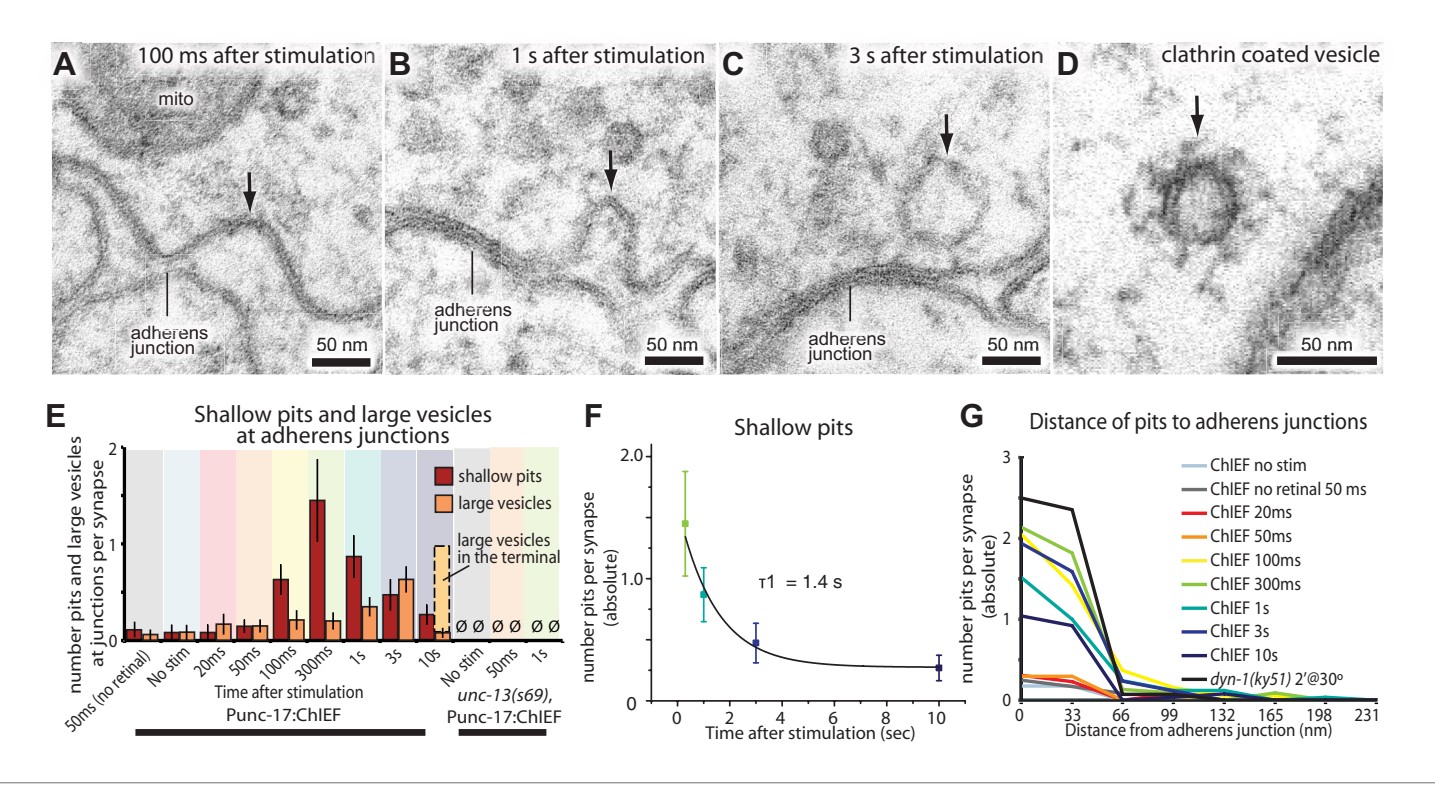

**Figure 6**. Endocytosis adjacent to adherens junctions is fast. (**A**) A shallow pit is present adjacent to the adherens junction (arrow) 100 ms after stimulation. (**B**) A deep pit at the adherens junction 1 s after stimulation. (**C**) A large clear vesicle near the adherens junction 3 s after stimulation. (**D**) Clathrin coats can be preserved through high-pressure freezing and freeze-substitution. An electron micrograph of a clathrin-coated vesicle in an endophilin mutant captured with the same fixation protocol and equipment. (**E**) Average number of shallow pits at adherens junctions (red) and large vesicles associated with adherens junctions (dark orange) per synapse. The decline of shallow pits is followed by an increase in large vesicles at adherens junctions. N values: no retinal 50 ms: 12 synapses; non-stimulated: 26 synapses; 20 ms: 12 synapses; 50 ms: 24 synapses; 100 ms: 19 synapses; 300 ms: 20 synapses; 1 s: 23 synapses; 3 s: 19 synapses; 10 s: 26 synapses. A synapse is defined as a collection of profiles containing the same dense projection. (**F**) Average number of shallow pits at adherens junctions per synapse. The time constant for the resolution of pits is 1.4 s. (**G**) Distance of shallow pits to adherens junctions after the stimulation. Average number per synapse is shown. The pits are next to adherens junctions. The standard error of the mean is shown in each graph.

The following source data and figure supplements are available for figure 6:

**Source data 1**. *Figure 6E,F*: Number of shallow pits at adherens junctions per synapse.

**Figure supplement 1**. Endocytosis takes place adjacent to adherens junctions.

Ceccarelli et al. (1972) stimulated at 2 Hz for 2 hr and fixed using ice-cold glutaraldehyde. Membrane invaginations with various neck sizes were observed: wide, narrow, and almost invisible. Horseradish peroxidase (*Ceccarelli et al., 1972*) and dextran (*Ceccarelli et al., 1973*) were taken up into these invaginations, suggesting that endocytosis takes place at the active zone. Given the presence of invaginations with a narrow neck, they speculated that exocytosis and endocytosis are coupled via the fusion pore. To support these conclusions with high temporal resolution images, they applied a single stimulus in the presence of 4-aminopyridine and 10 mM calcium and preserved structure using a freeze slammer (*Torri-Tarelli et al., 1985*). They did not observe exocytic figures accumulating between 2.5 ms and 10 ms, nor did they observe clathrin-coated vesicles at later time points as they expected based on the Heuser and Reese experiments, and concluded that vesicle fusion pores were reversing before collapse of the vesicles into the membrane (*Torri-Tarelli et al., 1985*).

These previous studies were performed in dissected preparations and K+ channels were blocked to maximize the exocytosis. Here, we developed a method to visualize the ultrastructure of endocytosis

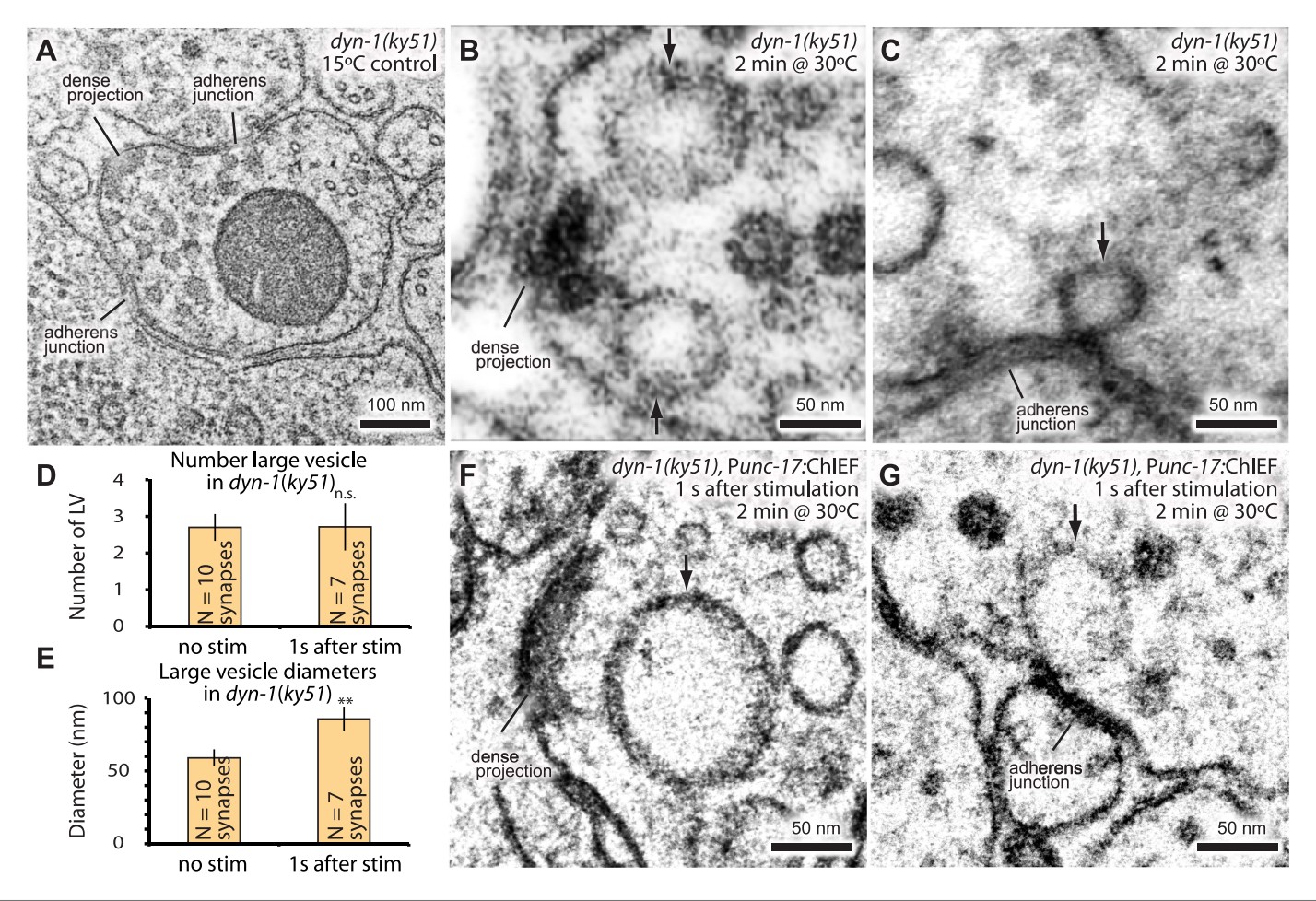

**Figure 7**. Dynamin is required for fast endocytosis. (**A**–**C**, **F** and **G**) Electron micrographs showing morphology of an acetylcholine motor neuron from the dynamin temperature-sensitive mutant *dyn-1(ky51)* at the permissive temperature (**A**) and at the restrictive temperature (30°C) for 2 min with no stimulation (**B** and **C**) or 1 s after stimulation (**F** and **G**). Large vesicles near the dense projection (**B** and **F**) and adherens junction (**C** and **G**) are stabilized only when the mutants were exposed to the restrictive temperature. Average number of large vesicles per synapse (**D**) and average diameter of large vesicles (**E**) in *dyn-1(ky51)* animals at the restrictive temperature. A single stimulus does not increase the number of endocytic structures, but the diameter of large vesicles increases.

The following source data and figure supplements are available for figure 7:

**Source data 1**. *Figure 7D*, Number of large vesicles; *Figure 7E*, Diameter of large vesicles.

**Figure supplement 1**. Dynamin is required for ultrafast endocytosis.

after a single stimulus in an intact *C. elegans* nervous system. Neurotransmission at *C. elegans* neuromuscular junctions is likely to be graded rather than driven by an action potential; however, light stimulation via channelrhodopsin has been used to study single evoked responses at these synapses (***Liu et al., 2009***). The goal was to look in a sample that was stimulated only once rather than after high frequency stimulations to avoid bulk endocytosis, and to look at very rapid time scales that would distinguish clathrin-mediated endocytosis from kiss-and-run endocytosis. We found that synaptic vesicle fusion takes place along the entire face of the synapse from the dense projection to the adherens junction. By contrast, endocytosis takes place at the edges of the active zone, either directly adjacent to the dense projection or laterally at the adherens junction. Endocytosis adjacent to the dense projection is completed 50 ms after a stimulus, and endocytosis at the adherens junction occurs about 1 s after stimulation. Both of these events are rapid, and do not fully support either kiss-and-run or clathrin-mediated mechanisms.

## Which vesicles fuse?

The 'active zone' is defined as the site of synaptic vesicle exocytosis, originally determined by ultrastructural experiments (*Couteaux and Pécot-Dechavassine, 1974*). We mapped the extent of the active zone at *C. elegans* neuromuscular junctions by directly visualizing fusing vesicles 20 ms after the initiation of a single stimulus, or by comparing changes in the distribution of docked vesicles in unstimulated synapses and synapses 50 ms after a stimulus. Vesicles fuse to the plasma membrane along the entire face of the synapse, from the dense projection to the adherens junctions. These results are not only important for determining the mode of endocytosis, but also for the identification of vesicle pools during exocytosis that have been previously defined by electrophysiological experiments.

The 'readily releasable pool' are vesicles that can be stimulated to fuse by a single action potential and these vesicles are determined by electrophysiological experiments (*Stevens and Tsujimoto, 1995*; *Rosenmund and Stevens, 1996*; *Schweizer and Ryan, 2006*). It is believed that the readily releasable pool corresponds to morphologically docked vesicles. These pools have been directly compared by combining dye-loading experiments and electrical stimulation in retinal bipolar cells (*Zenisek et al., 2000*). By using TIRF microscopy, the release ready vesicles were shown to be within ~10 nm of the surface (*Zenisek et al., 2000*). Docked vesicles can also be identified in electron micrographs. Careful comparisons of the pools for docked and for releasable vesicles demonstrated that they were of similar size (*Schikorski and Stevens, 1997*), and these were similar in size to the pool of vesicles that could be loaded with dye after stimulation by 40 action potentials (*Schikorski and Stevens, 2001*). Here, we reinforce the relationship between the readily releasable pool and docked vesicles by examining the ultrastructure after light-activated stimulation at the *C. elegans* neuromuscular junction. We found that only docked vesicles can fuse with the plasma membrane after a single stimulus. Vesicles that are tethered to the membrane, or within 30 nm of the membrane, or docked at 'ectopic' sites in the perisynaptic zone on the distal side of the synapse do not fuse in response to a stimulus.

A second important feature of the readily releasable pool is the refilling rate. Refilling of the readily releasable pool can be fit with two time constants in hippocampal neurons, retinal bipolar cells and the *C. elegans* neuromuscular junction. Hippocampal neurons refill the readily releasable pool with two time constants of 0.5 s and 4.3 s (*Pyott and Rosenmund, 2002*). Retinal bipolar cells refill the readily releasable pool with two time constants of 0.2 s and ~2.0 s (*Zenisek et al., 2000*). We also observe two time constants for the recovery of the readily releasable pool in *C. elegans:* a fast recovery of 0.1 s and a slow recovery of 2.2 s. What is the molecular basis for these two time constants observed in all three of these synaptic types?

The slow phase of recovery likely represents the refilling of the docked pool of vesicles. In retinal bipolar neurons, 'newcomers' can be observed to approach the plasma membrane and refill a stable docking site within about 2 s. Similarly, we find that docked vesicles recover with a time constant of 2.4 s. Thus, the firing rate of a synapse is in part limited by the speed of refilling the docked pool as previously postulated (*Liu and Tsien, 1995*; *Stevens and Tsujimoto, 1995*; *Rosenmund and Stevens, 1996*).

The fast time constant of 100 ms was not accompanied by ultrastructural changes at the *C. elegans* neuromuscular junction. *Zenisek et al. (2000)* found that newly docked vesicles required a 200 ms period to acquire fusion-competence. These authors argued that this was the time required for a molecular priming step. This step may correspond to the fast 100 ms time constant we observed for fast refilling of the readily-releasable pool in our electrophysiological study. About 60% of the docked vesicles fuse in response to a stimulus; these remaining docked vesicles may need to undergo a transition from an incompetent to a competent state (*Neher and Sakaba, 2008*; *Hosoi et al., 2009*). We cannot distinguish these two classes morphologically; the 'incompetent' vesicles would simply be vesicles that remain after the stimulus. These data would suggest that even some docked vesicles are not in a competent state for fusion.

## Coming in or going out?

We observed shallow pits appearing next to dense projections and next to adherens junctions immediately following a stimulus. One interpretation is that these structures are exocytic in nature, that is, the large vesicles are the result of fusions between synaptic vesicles, which then fuse to the plasma membrane. This model is described as 'compound fusion'(*Röhlich et al., 1971*; *Alvarez de Toledo and Fernandez, 1990*; *Matthews and Sterling, 2008*; *He et al., 2009*). For example, at the synaptic ribbon in retinal bipolar cells large vesicles and invaginations are observed after stimulation and are

thought to represent fusion between synaptic vesicles preceding exocytosis (*Matthews and Sterling, 2008*). However, the pits we observe are likely to be endocytic rather than exocytic intermediates for four reasons. First, we observe simple fusions along the membrane that are the size of synaptic vesicles, whereas the pits at the dense projection and adherens junction are larger than synaptic vesicles (*Figure 5A*). Second, the fusions of docked synaptic vesicle are coincident with postsynaptic currents, both are no longer present after 30 ms, whereas the invaginations at the dense projection are still present after postsynaptic currents have declined. Third, invaginations at these sites are stabilized in temperature-sensitive dynamin mutants suggesting these are endocytic in nature. Fourth, there is a temporal order to these structures suggesting a precursor-product relationship between them. It is more likely that the presence of invaginations at the plasma membrane, followed by the presence of large vesicles at these sites, and then the translocation of large vesicles into the synaptic terminal represents an influx of membrane rather than an efflux.

## Two endocytic sites?

The two sites we observe for endocytosis, at the dense projections and adherens junctions, may correspond to the two endocytic sites observed in other organisms. For example, in dynamin mutants in the *Drosophila* retina, trapped endocytic structures are found both adjacent to the dense projection and at lateral sites (*Koenig and Ikeda, 1996*). At the frog neuromuscular junction large indentations appear near the active zone 1 s after stimulation, in addition to small pits at lateral sites (*Miller and Heuser, 1984*). Optical imaging of the frog neuromuscular junction (*Richards et al., 2000*; *Gaffield et al., 2011*), the snake neuromuscular junctions (*Teng et al., 1999*), and in hippocampal cultures (*Vanden Berghe and Klingauf, 2006*) were also able to distinguish two sites of endocytosis. These two distinctive sites may direct recycling vesicles into different pools: recycling from the active zone may be directed into the cycling pool and those from the periactive zone could be directed to the reserve pool (*Richards et al., 2000*; *Vanden Berghe and Klingauf, 2006*; *Voglmaier and Edwards, 2007*).

## How fast is endocytosis?

Endocytosis at the dense projection occurs very rapidly: invaginations appear at 20 ms after the stimulation and these pits have resolved into large vesicles in less than 50 ms. This rate is extremely fast compared to measures of kiss-and-run endocytosis (~1 s; *Zhang et al., 2009*) and clathrin-mediated endocytosis (15–20 s [*Miller and Heuser, 1984*; *Jockusch et al., 2005*; *Balaji and Ryan, 2007*]) and we call this 'ultrafast endocytosis' to distinguish it from other models.

At the adherens junction endocytosis is not as fast as at the dense projection but is still very fast. Shallow pits peak at 300 ms and are mostly resolved by 3 s. However, the presence of some pits even 10 s after stimulation suggests that a slower mechanism is at work here. The time constant for clathrin-mediated endocytosis is reported to be ~15–20 s (*Miller and Heuser, 1984*; *Granseth et al., 2006*; *Balaji and Ryan, 2007*), which is in the range of the events we observe adjacent to the adherens junction. Even 15–20 s is extremely fast given the slow speed of clathrin assembly (*Kirchhausen, 2009*; *Cocucci et al., 2012*). It is possible that clathrin-mediated endocytosis at synapses is accelerated, possibly due to pre-formed clathrin cages at endocytic sites (*Wienisch and Klingauf, 2006*), or that clathrin is not involved in these fast forms of endocytosis.

## Clathrin-mediated or kiss-and run?

The distinguishing feature of clathrin-mediated endocytosis is that clathrin coats can be observed by electron microscopy. Coats are preserved when neurons are fixed with glutaraldehyde (*Heuser and Reese, 1973*; *Koenig and Ikeda, 1989*; *Takei et al., 1998*), and coated vesicles are preserved in the conditions used here. The absence of coats on endocytic pits is consistent with the absence of a genetic requirement for clathrin heavy chain at *C. elegans* neuromuscular junctions (*Sato et al., 2009*). However, making strong conclusions about the absence of a protein based on morphology in electron micrographs is highly risky. These results do not exclude the possibility that clathrin is required at synapses. In fact, synaptic vesicles are smaller in the temperature-sensitive allele of clathrin in *C. elegans* (*Sato et al., 2009*), suggesting that vesicles may pass through a clathrin-coated intermediate at the neuromuscular junction, perhaps during the resolution of endosomes.

Kiss-and-run endocytosis is thought to occur within about 1 s of stimulation (*Miller and Heuser, 1984*; *Zhang et al., 2009*). The appearance of endocytic structures within 50 ms after stimulation at the *C. elegans* neuromuscular junction is consistent with kiss-and-run endocytosis. However, there are two morphological features that suggest kiss-and-run is unlikely to account for endocytosis after

stimulation at the *C. elegans* neuromuscular junction: position and size. Kiss-and-run is predicted to occur at the site of exocytosis: the vesicle forms a fusion pore, which reverses itself to regenerate an empty vesicle. All exocytic intermediates captured at 20 ms after stimulation at the *C. elegans* neuromuscular junction have wide necks, suggesting that synaptic vesicles are collapsing into the membrane. Moreover, vesicles fuse throughout the active zone, but endocytosis takes place at the edges of the active zone. Although we cannot exclude the reversal of the fusion pore by some vesicles, a large amount of membrane must translocate to endocytic sites at the lateral margins of the synaptic membrane.

Not only the position, but the size of the endocytic structures does not fit a classical model for kiss-and-run endocytosis. The kiss-and-run model predicts that synaptic vesicles retain their integrity, that is, the vesicle is recovered intact (***Murthy and Stevens, 1998***). Thus, endocytic vesicles should be 30 nm in diameter. In fact, some of the vesicles recovered at dense projections were only slightly larger than a single vesicle. However, some endocytic structures adjacent to the dense projection are greater than twice the size of a synaptic vesicle. Moreover, the arrested endocytic structures in the dynamin mutants became even larger with prolonged synaptic activity. These very large trapped pits in dynamin mutants suggest that vesicles collapse into the plasma membrane and are retrieved via an endocytic pit.

Ultrafast endocytosis might require optogenetic stimulation methods to be observed: endocytosis is assayed in this preparation in an undissected animal. Neurons in intact nervous systems are bundled together, thus, in intact nervous systems spatial constraints may accelerate synaptic vesicle endocytosis even during low frequency activity.

## Materials and methods

### Strains

N2 Bristol was used as the wild type in all experiments. All strains were maintained at 22°C on standard NGM media seeded with OP50 except for *dyn-1(ky51)*, which was kept at 15°C. EG5793 *oxSi91[Punc-17::ChIEF::mCherry::unc-54UTR; unc-119*(+)] *II* was generated using a MosSCI direct insertion protocol (***Frøkjaer-Jensen et al., 2008***). Although MosSCI usually generates single copy inserts, the *oxSi91* transgene was a rare event specifically selected for high expression and likely represent the insertion of multiple copies. pQL12 plasmid was injected into EG4322 *ttTi5605; unc-119(ed3)* worms. The injection mix consisted of 25 ng/ml targeting plasmid (pQL12), 10 ng/ml pCFJ103 (*Ppie-1::mosase::pie-1 UTR*), 5 ng/ml pGH8 (*Prab-3::mCherry*), 5 ng/ml pCFJ104 (*Pmyo-3::mCherry*) and 2.5 ng/ml pCFJ90 (*Pmyo-2::mCherry*). Injected worms were individually transferred to standard NGM plates at room temperature and allowed to exhaust the food source. Once starved, plates containing transgenic lines were screened for insertion events on a fluorescence dissection microscope based on wild-type movement but lack of fluorescent coinjection markers. Several independent lines were obtained and EG5793 was selected for its strong mCherry expression in the nervous system. The strong mCherry expression suggests that a multi-copy transgene was inserted.

Other strains used in this study are:

EG5899 acr-16(ok789) V;oxSi93[Punc-17::ChIEF::mCherry; unc-119+] II
EG5183 unc-13(s69) I; oxIs364[Punc-17::channelrhodopsin-2::mCherry; lin-15+; Litmus38i].
CX51 dyn-1(ky51ts) X.

### Retinal feeding

Transgenic worms were grown in the presence of all-trans retinal 16–48 hr before experiments. All trans-retinal (Sigma, St. Louis, MO) was dissolved in ethanol to make 100 mM stock solution and stored at −20°C in the dark. NGM plates were seeded with 4 µl 100 mM retinal stock solution mixed with 250 µl OP50 per 50-mm plate. Seeded retinal plates were kept in the dark at 4°C for up to a week. Young adult transgenic worms were transferred from regular plates to freshly seeded retinal plates in the dark at room temperature.

### Electrophysiology

Young adult hermaphrodite animals were used for electrophysiological analysis. Tonic and evoked postsynaptic currents (mPSCs and ePSCs) at the neuromuscular junction were recorded as previously described (***Richmond et al., 1999***; ***Liu et al., 2009***). Evoked responses are not observed in

non-transgenic animals with exogenously added retinal or transgenic animals expressing channelrhodopsin in absence of trans-retinal (*Nagel et al., 2005*; *Liewald et al., 2008*). In brief, an animal was immobilized on a Sylgard-coated glass coverslip by applying a cyanoacrylate adhesive along the dorsal side. A longitudinal incision was made in the dorsolateral region. The cuticle flap was folded back and glued to the coverslip, exposing the ventral nerve cord and two adjacent muscle quadrants. An upright microscope (Axioskop; Carl Zeiss, Inc., Jena, Germany) equipped with a 40× water immersion lens and 15× eyepieces was used for viewing the preparation. Borosilicate glass pipettes with a tip resistance of ~ 3–5 MΩ were used as electrodes for voltage clamping. The classical whole-cell configuration was obtained by rupturing the patch membrane of a gigaohm seal formed between the recording electrode and a right medial body muscle two cells anterior to the vulva. The cell was voltage clamped at −60 mV to record mPSCs and ePSCs. Postsynaptic currents were amplified (EPC10; HEKA, Bellmore, NY) and acquired with Patchmaster software (HEKA). Data were sampled at a rate of 10 kHz after filtering at 2 kHz. The recording pipette solution contained the following: 120 mM KCl, 20 mM KOH, 5 mM TES, 0.25 mM $CaCl_2$, 4 mM $MgCl_2$, 36 mM sucrose, 5 mM EGTA, and 4 mM $Na_2ATP$; adjusted to pH 7.2 with KOH and osmolarity to 310–320 mOsm with sucrose. The standard external solution included the following: 150 mM NaCl, 5 mM KCl, 5 mM $CaCl_2$, 1 mM $MgCl_2$, 5 mM sucrose, 10 mM glucose, and 15 mM HEPES; adjusted to pH 7.35 with NaOH and an osmolarity of 330–340 mOsm. Amplitude and frequency of mPSCs were analyzed using MiniAnalysis (Synaptosoft, Decatur, GA). A detection threshold of 10 pA was used in initial automatic analysis, followed by visual inspections to include missed events (≥5 pA) and to exclude false events resulting from baseline fluctuations. Amplitudes of ePSCs were measured with Patchmaster (HEKA) and analyzed with Fitmaster. The amplitude of the largest peak of ePSCs from each experiment was used for statistical analysis. Data were imported into Origin, version 7.5 (OriginLab, Northampton, MA), for graphing and statistical analysis. Unpaired *t*-test was used for statistical comparisons. A p value of <0.05 was considered statistically significant.

## Illumination system for electrophysiology

The light source was an AttoArc 2 HBO 100W system (Zeiss) with a mercury arc bulb (Osram, Munich, Germany). A Zeiss bandpass filter set was used to excite channelrhodopsin (at 450–490 nm. Light intensity was controlled by adjusting dialing the power supply. The light on/off switch was controlled by a Uniblitz VS25 shutter and a VCM-D1 driver (Uniblitz, Rochester, NY). Light-activated currents in ChIEF transgenic animals were evoked by shining light (10 mW/mm$^2$ unless specified) onto the dissected worm preparation on a Zeiss Axioskop microscope, equipped with a 40× water-immersion objective and 15× eyepieces. Rapid shutter opening and closing was triggered by TTL signals from the HEKA EPC-10 amplifier. A liquid light guide (Sutter, Novato, CA) was installed between the Axioskop microscope and the shutter to avoid mechanical vibrations during repetitive stimulations. Light intensity was measured with a Coherent FieldMax II-TOP Laser Power/Energy Meter.

## Paired-pulse protocol

A single 10 ms light pulse was followed by a second 10 ms pulse after a variable delay for each trial. These inter-pulse intervals ranged from 50 ms to 15 s. Between consecutive trials there was >30 s of darkness to allow for complete channel recovery. The ratios of the second current peak to the first current peak were calculated and plotted relative to the recovery intervals.

## Modifications of the Leica EMpact2

Freezing was conducted using a Leica EMpact2 (Leica, Wetzlar, Germany), which does not include the rapid transfer system (RTS) attachment. To create a light path, the specimen pod (#16707829; Leica) and the manual loading bayonet for the EMpact2 (#16707828; Leica) were modified. The anvil mounting screw in the specimen pod was drilled out and machined with a countersink on the top of the screw (*Figure 1C*). The black diamond anvil was removed and replaced with a sapphire disk (#E2.08; SwissJewel) with a diameter of 2.08 mm and a thickness of 1 mm (*Figure 1B,C*). The sapphire anvil was glued into the countersink of the screw using two-part epoxy (#14250; Devcon, Danvers, MA) (*Figure 1C*).

A 2-mm hole was drilled through the middle of the bayonet (*Figure 1C*). A 3-mm blue light-emitting diode (LED) was inserted into the bayonet just behind the specimen pod (*Figure 1C*). For initial experiments, a power supply for the LED was constructed using a 9 V battery, a TIP110 transistor and a 200 kΩ current limiting resistor. This was connected to the LED using sub micro twisted servo wire (Radical RC, Dayton, OH, www.radicalrc.com, W3ST-34JR). The transistor base was connected to an

analog output channel of a USB multifunction DAQ device (NI-USB6008, National Instruments, Austin, TX, www.ni.com). The DAQ device was driven by a custom Labview program (available on request) which could monitor an accelerometer mounted to the bayonet holder and update the LED output level every 7–8 ms. In later experiments, we constructed a dedicated device based on an Arduino Uno microcontroller board (www.arduino.cc/) driven by custom firmware. For these experiments, we changed from the 3-mm LED to a 3.2 mm diameter high brightness LED (LZ1-00B200; LED Engin, www.ledengin.com/), fitted into the bayonet with a custom-machined adapter. The LED is powered using a dedicated constant current LED power module (7023-D-E-1000; Luxdrive, www.luxdrive.com/), which is controlled by the microcontroller. The microcontroller triggers the freezing of the sample by communicating with the EMpact-2 via an RS232 serial communication channel. The device can be reproduced by Marine Reef International, Newport Beach, CA.

## Stimulation and high-pressure freezing

For inducing neurotransmission, 10 young wild-type or *unc-13(s69)* animals expressing ChIEF in the acetylcholine neurons (*Punc-17::ChIEF::mCherry*) were mounted onto a membrane carrier with a depth of 100 µm (#16707898; Leica) and stimulated with a single 20 ms pulse of blue light (488 nm). To maximize the absorbance of light by specimens, 20% BSA (A3059-10 G; Sigma-Aldrich) was used as a cryoprotectant instead of bacteria. We set the light stimulation device so that a light pulse was triggered at 20 ms, 30 ms, 50 ms, 100 ms, 300 ms, 1 s, 3 s, and 10 s before freezing. The actual time when the light pulse was applied relative to freezing was calculated based on the movement recording from the accelerometer and the output of the light pulse recorded by the computer (see the next section).

## Calculations for the interval between stimulation and freezing

The light stimulus was correlated post facto with the freeze. To calculate the actual interval between stimulation and freezing, we summed the time between light stimulation and pressure application and the time between pressure application and freezing. The interval between the pressure and freezing can be determined based on values recorded by the high-pressure freezer, and it was consistently 8 ms. To calculate the interval between light and pressure, we needed to monitor when the application of pressure is initiated after sending the 'start' signal to the high-pressure freezer. To measure when pressure is applied, we mounted an accelerometer on the specimen bayonet. The bayonet jolts when the pressure is applied to the specimen. We could then record the interval between the application of light and pressure.

We programmed our light device so that the desired interval between stimulation and freezing could be achieved by simply modifying the timing of the output signal to initiate the freezing process. Based on the accelerometer reading, the pressure increase occurs around 170 ms after sending out the 'start' signal. The 170 ms delay is required for the operating system of the freezer to communicate with the pressure firing switch. Since the freezing initiates 8 ms after the pressure application, 178 ms is required for specimens to freeze after sending out the 'start' signal. Thus, to freeze a sample 1 s after stimulation, one needs to activate the high pressure freezer 822 ms after the light stimulus; to freeze a sample 100 ms after stimulation, one needs to activate the high pressure freezer 78 ms 'before' the light. However, in our experiments, we found that the 170 ms delay is not consistent—it varies by ±20 ms. Thus, the actual time between light stimulation and freezing was calculated post hoc from the accelerometer read-out vs when the light was stimulated. Unfortunately, for very short time points, the light was often flashed 'after' the sample was frozen because of fluctuations in the pressure time point.

## Post-stimulation electron microscopy

After freezing, the specimens were transferred to the cryovial containing 2% glutaraldehyde (#16530; EMS, Hatfield, PA) in anhydrous acetone (#RT10016; EMS) under liquid nitrogen. The use of glutaraldehyde as a primary fixative was necessary for preserving clathrin coats when cryo-methods are employed (Forostyan, Watanabe, Jorgensen, unpublished). The freeze-substitution was carried out in a Leica Automatic Freeze Substitution unit (AFS2; Leica) with the following program: 48 hr at −90°C, 5°C/hr to −60°C, 6 hr at −60°C, 5°C/hr to −20°C, 16 hr at −20°C, and 10°C/hr to 20°C. The glutaraldehyde was removed and the sample rinsed with acetone six times when the program reached −60°C. The fixative was then changed to 1% osmium tetroxide (#RT19130; EMS) + 0.1% uranyl acetate (#RT22400; EMS) at −60°C. The vials were swirled at least twice a day to promote the diffusion of fixatives into tissues. After the completion of the program, the fixative was removed using six acetone washes. The infiltration of plastic, araldite-epon (#18,028; Ted Pella, Redding, CA), was carried out on a nutator with

a gradual change in the concentration of plastic (30%, 70%, and 90%). The next day, the specimen was transferred into the cap of a BEEM capsule (#70010-B; EMS) containing 100% fresh resin. Following three changes of resin over 6 hr, the plastic was polymerized at 60°C for 48 hr.

## Electron microscopy

We imaged and analyzed three animals from each condition blind. 250–300 contiguous sections were cut using a microtome (UC6; Leica) from each condition and collected onto formvar–coated (#RT15820; EMS) grids (#1GC12H; Ted Pella). The sections were stained with 2.5% uranyl acetate in 70% methanol for four minutes prior to imaging. Sections were imaged on a Hitachi H-7100 electron microscope equipped with a Gatan digital camera (Gatan, Orius, Pleasanton, CA).

## Electron tomography

For electron tomography imaging, grids were first coated with 0.5% pioloform, and then carbon was sputtered on top of the film for ~10 s. Using a microtome as described above, arrays of 200 nm thick sections were collected onto the carbon-coated grids. The sections were imaged on a FEI Tecnai G2 F20 fitted with an Eagle HR CCD camera. The accelerating voltage was set at 200 keV. A tilt series of ±65° was collected from each section. The tomograms were computed and reconstructed from the tilt series using IMOD.

## Morphological analysis

We developed a macro for ImageJ and a Matlab program for morphological analysis of synapses (Watanabe, Davis, and Jorgensen, unpublished). From each axon profile containing a slice of the dense projection or one section adjacent to the dense projection, the locations of vesicles, plasma membrane, dense projection, and adherens junctions, as well as the diameters of vesicles, were annotated and exported as a text file. Data were analyzed as per 'profile', that is for synaptic vesicle docking or fusions, for which many events can be detected along the membrane. Alternatively, data were analyzed per 'synapse', that is a reconstructed region of a synapse around the dense projection, for which only one or two events occur such invaginations or the presence of large vesicles. Vesicles were categorized into three types depending on their appearance: synaptic vesicles, dense core vesicles, and large vesicles. Synaptic vesicles have a diameter of ~30 nm and are most abundant in the terminals. Synaptic vesicles that are in the physical contact with plasma membrane are categorized as docked vesicles. Other vesicles are close to the membrane (<30 nm) but are not in contact with the membrane. Such vesicles with a physical tether are categorized as 'tethered', and such vesicles with no obvious tether are categorized as the '30 nm pool'. Note that a combination of both pools was previously defined as the 'tethered pool' (*Broadie et al., 1995*; *Toonen et al., 2006*; *Hammarlund et al., 2007*; *Siksou et al., 2009*; *Gracheva et al., 2010*). Dense core vesicle can be distinguished by the dark appearance of the vesicle core, and their diameter is typically around 40 nm. Large vesicles are clear-core vesicles that are larger than 35 nm in diameter. Then, these text files were imported into the Matlab analysis program. In this program, the distances from each vesicle to the nearest edge of dense projection, to the closest plasma membrane, and to the adherens junctions were calculated. The total number of vesicles in the terminal was normalized based on the average area of the synaptic profile to account for the differences in the size of each terminal. The diameter of synaptic varicosities can vary greatly, although the density of vesicles within the varicosity does not. This variation can lead to irrelevant changes in the number of synaptic vesicles per profile from time point to time point. Thus, each value for the number of synaptic vesicles in a profile (*Figure 4D–F*) was normalized to the average profile area. Values thus really represent the number of synaptic vesicles in a typical synaptic varicosity with a cross-section of 60,700 nm$^2$. Similarly, the numbers of vesicles docked or tethered along the active zone (*Figure 2H,J* and *Figure 3D,E*) were normalized by the average length of active zones from the dense projection to the flanking adherens junctions. Values thus really represent the number of synaptic vesicles along a typical active zone membrane of 670 nm.

## 3D model generation

The structural features were segmented from serial section electron micrographs and a 200 nm tomogram using a pen tablet display (Wacom Cintiq 21UX, Vancouver, WA) and Amira (version 4.1.2). Synaptic vesicles were created using the landmark tools. Plasma membrane was traced in a way that

does not leave holes and gaps between sections, thus the thickness of plasma membranes does not reflect the actual dimensions of bilayers.

## Membrane calculations

To determine if membrane added to the surface is balanced by membrane recovered by recycling we compared the amount of membrane added by exocytosis to that removed by endocytosis. Moreover, we made this comparison at two time points: 300 ms and 1 s. These time points were selected because endocytic structures can be quantified at both the dense projection and at the adherens junction at these time points. In brief, we find that the membrane added to the surface is similar to the area removed at both of these time points.

To calculate the total amount of recycled membrane, we made the following three assumptions. First, we assumed that our partial reconstructions would provide a reasonable sampling of exocytic and endocytic events at a synapse. Vesicles and endocytic figures were only analyzed in profiles containing a dense projection. These profiles are cross sections of a single synapse that include active zone membrane on two sides from the dense projection to the flanking adherens junctions. The dense projection spans approximately four sections (*Hammarlund et al., 2007*) and these sections encompass about one third of the vesicles found at a synapse. Thus, our reconstructions sample a significant fraction of vesicles at each synapse. On the other hand, it is likely that exocytic and endocytic events are occurring at the active zone in adjacent sections, leading to a systematic undercounting of recycling events per synapse. We confirmed the presence of exocytic and endocytic pits in profiles not containing a dense projection by examining adjacent sections. Endocytic pits in these sections were also strictly associated with adherens junctions but were not included in our calculations. Because these endocytic events are likely driven by local exocytosis, it is possible that exclusion of these domains does not bias our calculations.

Second, we assumed that large vesicles but not pits are the best estimate for membrane recovery at the dense projection (at 300 ms and 1 s time points). Specifically, pits have a short duration at the dense projection but large vesicles are slowly translocated into the terminal. Pits are not observed at dense projections after the 30 ms time point and are therefore resolved by the 50 ms time point.

Third, we assumed that pits (as well as large vesicles) are a good estimate of endocytosis at the adherens junctions since they are more stable than those at the dense projection. Shallow pits peak at 300 ms at the adherens junction, and decline with a time constant of 1.4 s. Conversely, large vesicles accumulate between 1 and 3 s. Deep pits are only occasionally observed in the 1 s time point and must resolve into large vesicles rapidly. We assumed that the shallow pits observed at the adherens junctions will be resolved as a large vesicle with a diameter of ~43 nm. Thus, pits and large vesicles can be counted as endocytic events at the adherens junction.

We did not count large vesicles accumulating in the terminal in our calculations because first, their provenance is unknown and second, they are likely to be drifting laterally or resolving so that their presence may be unreliable. However, because large vesicles are accumulating in the terminal at 100 ms, our values for membrane recovered by endocytosis is likely to be an underestimate at both the 300 ms and 1 s time points.

For the 300 ms time point we analyzed 22 synapses (where a synapse is the sum of the ~four profiles containing a dense projection). Since six to seven vesicles are typically released per synapse, then the area from 132–154 vesicles (~143 SVs) was added to the surface. Given that the surface area of single synaptic vesicle of 29.3 nm is 2700 nm$^2$ ($4\pi r^2$), then the total amount of membrane will be 356,000–416,000 nm$^2$ (a mean of 143 synaptic vesicles or 386,000 nm$^2$).

We then calculated the number of synaptic vesicle equivalents that we observed endocytosed at the 300 ms time points. At the 300 ms time point, there were 14 large vesicles at dense projections. Based on the time course of their decline, ~47% of the large vesicles (~12 large vesicles) would have diffused away from the dense projection by this time (*Figure 5B*), so a total of ~26 large vesicles were likely internalized at dense projections. The average diameter of 18 of these vesicles (70% of the vesicle at the dense projection) was ~37 nm (surface area = 4300 nm$^2$), and the remaining eight had an average diameter of 46 nm (surface area = 6600 nm$^2$) (*Figure 5I*). Thus, the total surface area of membrane recovered at dense projections is 130,200 nm$^2$. We also observed 29 shallow pits at adherens junctions. Assuming that each endocytic structure resolves into a large vesicle with a diameter of 43.2 nm (surface area = 6000 nm$^2$) (*Figure 6—figure supplement 1G*), about 174,000 nm$^2$ of membrane was retrieved at adherens junctions. Thus, for 356,00–416,000 nm$^2$ released, 304,200 nm$^2$ were recovered

or projected to recover based on the number of endocytic structures present 300 ms after stimulation. Thus, of the 143 synaptic vesicles that fuse, we can account for 112 synaptic vesicle equivalents by the observed endocytic structures at the 300 ms time point. The total amount of membrane exocytosed and endocytosed is roughly equal, suggesting that most of the endocytic structures are likely present by 300 ms after stimulation.

Calculations from the 1 s time point yielded similar results. We analyzed a total of 23 synapses 1 s after stimulation. Since four to six vesicles are typically released per synapse, the area from 138–161 vesicles (370,000–435,000 nm$^2$) was added to the surface. At the 1 s time point, there were 25 large vesicles at dense projections. Based on the data from 50 ms, ~50% of the large vesicles (~25 large vesicles) would have diffused away from the dense projection by this time (*Figure 5B*), so a total of ~50 large vesicles were likely internalized at dense projections. The diameter of 35 of these vesicles (70% of the vesicles at the dense projection) was ~37 nm (surface area = 4300 nm$^2$), and 15 had a diameter of 46 nm (surface area = 6600 nm$^2$). Thus, the total membrane recovered at dense projections is 249,500 nm$^2$. We also observed 20 shallow pits and four large vesicles at adherens junctions. We assume that all of these will resolve into large vesicles of 43.2 nm in diameter like the deep pits observed at the adherens junctions later. Each of these vesicles removes 6000 nm$^2$ ($4\pi r^2$) from the surface, so the area recovered or projected to recover at adherens junctions is 140,000 nm$^2$. With a range of 370,000–435,000 nm$^2$, roughly 150 vesicles were exocytosed, and about 390,000 nm$^2$ or 144 vesicle equivalents were recycled, again suggesting that all endocytic structures are likely formed very rapidly—in less than a second.

## Statistics

For complete data and p-values, please see: *Figure 2—source data 1*, *Figure 3—source data 1*, *Figure 4—source data 1*, *Figure 5—source data 1*, *Figure 6—source data 1* and *Figure 7—source data 1*.

In *Figures 2 and 3*, the number of fusing, docked or tethered vesicles is normalized to the length of the typical active zone or perisynaptic zone. Distributions in *Figures 2F and 3G* are not normalized but are absolute values. For *Figure 4*, docked vesicles, tethered vesicles and total vesicles are normalized to a typical profile area, and docked and tethered vesicles are not separated into active or perisynaptic zones. In *Figures 5–7*, serial sections containing a dense projection were considered 'a synapse', typically about four-six sections. Large vesicles were averaged per synapse. p values were determined in Mann-Whitney U tests because of the sample size and skewed distribution of the data. The confidence level was set at 95%. We applied the Bonferroni correction for multiple comparisons, and thus the confidence level is 0.007 for the P*unc-17:ChIEF* experiment in *Figure 4D–F*. Gray shading in the p-value columns of the source data files indicate that the observed difference is statistically significant.

## Acknowledgements

We would like to thank the Grass Foundation, the Marine Biological Laboratory at Woods Hole and Leica Inc. for providing us space and equipment necessary to perform the freezing experiments. We would like to thank Roger Y Tsien for providing us a construct for ChIEF, Robert J Hobson for critical reading of the manuscript, Carl Ebeling for advice on statistics, and Stefan Eimer and Alexander Gottschalk for communicating results before publication. EMJ is an Investigator of the Howard Hughes Medical Institute.

## Additional information

### Funding

| Funder | Grant reference number | Author |
| --- | --- | --- |
| National Institutes of Health | NS034307 | Erik M Jorgensen |
| National Science Foundation | 0920069 | Erik M Jorgensen |
| Howard Hughes Medical Institute | | Erik M Jorgensen |
| Marine Biological Laboratory (The Dart Scholars Program) | | Erik M Jorgensen |

The funders had no role in study design, data collection and interpretation, or the decision to submit the work for publication.

## Author contributions

SW, Conception and design, Acquisition of data, Analysis and interpretation of data, Drafting or revising the article; QL, Acquisition of data, Analysis and interpretation of data; MWD, Construction of hardware and software, Editing and approval of final manuscript, Conception and design; GH, Contributed strains and reagents, Design of experiments, Editing and approval of final manuscript; NT, Approval of final manuscript, Acquisition of data, Analysis and interpretation of data; NBJ, Construction of experimental apparatus, Approval of final manuscript, Acquisition of data; EMJ, Conception and design, Drafting or revising the article

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
