## [Decision Letter]

[Editors’ note: although it is not typical of the review process at *eLife*, in this case the editors decided to include the reviews in their entirety for the authors’ consideration as they prepared their revised submission.]

Thank you for sending your work entitled “Two modes of ultrafast endocytosis at synapses” for consideration at *eLife*. Your article has been favorably evaluated by Eve Marder (Senior editor) and 3 reviewers, two of whom, Tim Ryan and Volker Haucke, have agreed to reveal their identity.

The Senior editor and the three reviewers discussed their comments before we reached this decision. All of the reviewers were enthusiastic about your manuscript, and all felt that it should be published by *eLife*. All of the reviewers read your manuscript with care, and each has specific comments and suggestions for revision that we would like you to consider carefully. The reviewers all felt that you should revise your manuscript as you see appropriate, using their comments as indications of issues that arise in the minds of interested and careful readers. We assume that their comments will be useful to you and therefore we are including the three reviews in their entirety, as we are leaving it up to you the revisions you might wish to implement.

Reviewer 1:

This is a remarkable tour de force paper. The combination of high-pressure freeze with temporal control of stimulation in a system that is amenable to genetic manipulations has been a long-sought goal of molecular & cellular synaptic neuroscience. Our thinking in this field has been shaped by the classic papers of John Heuser, Tom Reese and Bruno Ceccarelli but this close up ultrastructural view of stimulation changes presynaptic ultrastructure has not advanced much since the mid-80s.

The work described here contains a lovely and comprehensive description of exocytosis: the ability to capture classic omega-figures, docked and tethered vesicles provides a convincing framework for looking at vesicle dynamics and pool refilling. The close correspondence of the recovery from paired pulse depression and the docked pool re-population dynamics argues strongly that the authors are able to follow the correct physiological process.

These exocytosis results are important for two reasons: they provide a nice framework for future molecular analyses and they validate the approach so that one should believe what they find in the endocytic phase. None of what they describe could have been expected. There is precedence for the larger vesicle formation near the dense projection as Heuser reported something similar (albeit definitions of active zones and dense projections are somewhat different); however the speed of the reformation of the smaller endocytic events are nothing one would have predicted. This paper will be of high interest to the field – it is well written and easy to understand – one quickly gets the main message that endocytosis is not happening as one would have imagined. This result will be difficult to accept for some but I see no flaws in the experiments or interpretations.

I am a little disappointed with the level of analysis of the *ky* dynamin mutant, but it is well known that there is really no equivalent to either the *Drosophila shibire* or mammalian k44A mutants in worms. It would have been nice to see ChIEF stimulated synapses in the mutants however. The fact that the analysis of the amount of exo matches the analysis for the amount of endo strongly suggests that the authors are tracking the right stuff here.

Another note of caution would be to discuss the fact that unlike in the previously documented systems, one does not really have any idea what a “stimulus” means here. The ChIEF is meant to mimic the phasic type of stimuli one gets out of an action potential, but this system does not have action potentials. This is the only real weakness and prevents one from fully embracing these results. However I consider this a necessary evil at this point and I think the paper should be published in *eLife*.

Reviewer 2:

The authors describe a time resolved (resolution ∼10's of milliseconds) ultrastructural analysis (by high-pressure freeze EM) of SV exocytosis and endocytosis at the *C. elegans* cholinergic NMJ. This study is a tour de force that will be referred to by experts in the field for many years to come. The authors develop a modern version of the Heuser slam freeze protocol, utilizing photo-evoked ACh release. Samples are frozen as quickly as 20ms following photostimulation, allowing for a detailed temporal analysis of changes in synapse ultrastructure. A critical aspect of this study is the use of high pressure freezing to preserve membrane ultrastructure, which gives the most accurate description of SV docking. Using this experimental design Watanabe and colleagues analyze where and when SV exocytosis and endocytosis happen with high spatial and temporal resolution. The location of SV fusions is determined by two criteria. First, within 20ms of stimulation, they see shallow invaginations of the active zone plasma membrane, which have an average diameter indistinguishable from SVs. These structures are distributed throughout the length of the active zone (extending roughly 400 nm radially from the dense projection), are gone by 50ms following stimulation (when evoked synaptic charge transfer is completed). The authors conclude that these shallow pits correspond to evoked SV fusions. Second, they compare the spatial pattern of SV docking before and after stimulation. In this manner, they observe a loss of 10-20 docked SVs within 20ms, which recovers in the next several seconds (tau ∼2 s). The authors also describe endocytic events, which are defined as shallow invaginations with diameters larger than SVs, and the subsequent appearance of larger diameter vesicles. These large diameter pits were observed at 20ms, persist for ∼1 second after stimulation, and their persistence is exaggerated by mutations inactivating dynamin. Following the appearance of the endocytic structures, larger diameter vesicles are observed, which are interpreted as the scission products of the endocytic pits. By these criteria, endocytic events occur in two zones: adjacent to either the dense projection or the adherens junction (which forms the lateral boundary of the active zone). Both forms of endocytosis occur rapidly, proximal and distal events peaking at 30ms and 300ms respectively. These experiments (and the resulting data) are elegant, and the authors’ interpretations and conclusions are convincing. Thus, I believe that this study will be of great interest to the broad community of synaptic cell biologists. I have a few minor comments, which the authors may wish to consider prior to publication.

Specific comments:

1) Are exocytic pits (i.e., diameter <30 nm) eliminated in UNC-13 mutants? I am assuming that the data in Figure 5 refer only to large diameter pits.

2) Is there a spatial pattern to the recovery of docked SVs following stimulation? The authors very nicely show that docked SVs recover with a time constant of ∼2 seconds (Figure 4). It would be very interesting to know the spatial location of the newly docked SVs. Do proximal and distal docked SVs appear with distinct kinetics? You have all of these data, so it would be great to analyze the spatial pattern of docking recovery.

3) I did not understand how the SV docking data were plotted in the inset of Figure 2J. Does the flesh tone shading indicate the number of docked SVs at each radial position? Does the inset illustrate how the radial positions were segmented into bins? If so, the shading doesn't clearly illustrate the binning to me (or my visual acuity has declined...).

4) How many total exocytic pits were analyzed in Figure 2J? It would make it easier to interpret these data if the total number of pits were reported.

5) I found it hard to discern the difference in shading between docked SVs and exocytic pits in Figure 2.

6) I found some of the p-values shown in the table in Supplementary file 1 surprising. For example, p=0.01 for total SVs in control (32) versus after stimulation (31). Is this p-value correct?

7) Is the total number of tethered SVs sufficient to account for the fast component of recovery (tau = 100 ms) in paired pulse experiments?

8) The data in Figure 3 indicate that docked SVs <600 nm from the dense projection are lost following stimulation. Where is the adherens junction in this diagram? I found this confusing because I thought that the peri-synaptic SVs are ∼400 nm from the DP and are not supposed to undergo evoked fusion.

Reviewer 3:

Watanabe et al. report here on the development of a system that capitalizes on optogenetic stimulation combined with rapid high pressure freezing to visualize synaptic vesicle (SV) exo-endocytosis at *C. elegans* NMJs within tens of milliseconds. Using this system they report that SVs recycling occurs by two ultrafast mechanisms of endocytosis, neither of which appears to conform to the previously postulated kiss-and-run or clathrin-dependent modes of SV retrieval.

Jorgensen and colleagues apply single light stimuli to transgenic worms expressing a non-desensitizing variant of channel rhodopsin and then use rapid high pressure freezing to visualize intermediates of SV cycling. They show that docked SVs are preferably consumed during single light stimuli, which release an equivalent of 4–6 SVs, i.e., about half of the docked SV pool (in these only partially reconstructed synapses). Loss of docked SVs is followed by the rapid appearance of shallow invaginations and large vesicles within 20-50 ms post-stimulation. Similar structures appear at distant adheres junctions within about 300 ms. The time course of appearance and consumption suggests a precursor-product relationship between invaginations and large vesicles. Further, disappearance of endocytic intermediates correlates with the replenishment of docked SVs and with the refilling of the RRP. Lastly, it is shown that resolution of enlarged invaginations into vesicles requires dynamin activity. Based on these results it is proposed that SV endocytosis in *C. elegans* involves two ultrafast mechanisms of membrane retrieval that occur at dense projections and at distant adherens junctions.

This is an excellent paper that likely will cause some stir in the community. The combined optogenetic/high-pressure freezing approach used here reveals unexpected and fundamental properties of SV exo-endocytosis and suggests that SV recycling may occur by mechanisms distinct from hitherto favored models. I have a few questions that would need to be addressed in a revised version of this manuscript before publication in *eLife*:

It would be important to provide further evidence that the invaginations seen post-stimulation (Figure 5) really do represent endocytic rather than exocytic structures. Do these contain dynamin (i.e., by immunogold or by correlative SRLM/ EM)?

Whether or not invaginations near dense projections vs. adherens junctions really represent two mechanisms of endocytosis in my opinion cannot be concluded from the data. What rather appears different is their time course. How do the authors envision the function of AJs as endocytic sites? Is it conceivable that common structural determinants underlie ultrafast endocytosis at dense projections and AJs? This potential caveat should also be reflected in the title.

---

## [Author Response]

*Reviewer 1*:

We thank the reviewer for a thorough read of our manuscript and for positive feedback.

*I am a little disappointed with the level of analysis of the ky dynamin mutant, but it is well known that there is really no equivalent to either the* Drosophila shibire *or mammalian k44A mutants in worms. It would have been nice to see ChIEF stimulated synapses in the mutants however. The fact that the analysis of the amount of exo matches the analysis for the amount of endo strongly suggests that the authors are tracking the right stuff here*.

The reviewer is asking us to demonstrate that dynamin actually functions to sever the endocytic intermediates from the surface of the membrane. In response to this criticism, we stimulated dynamin mutants and froze them 1 s later. The number of trapped structures did not increase (Figure 7), but they became ∼1.5 times larger (Figure 7). This increase in diameter corresponds to the surface area of ∼four synaptic vesicles per each structure, or 12 synaptic vesicles worth of membrane from each synapse. This number is roughly equal to the number of vesicles expected to fuse in response to a single stimulus. These results further confirmed that dense projections and adherens junctions are the endocytic sites in *C. elegans* neuromuscular junctions and that dynamin is involved in cleaving the neck of these endocytic structures. We included these new results in the revised manuscript (new text starting “Endocytic structures are often cleaved from the surface by dynamin (36; 71). To test if dynamin is also required to resolve ultrafast endocytic intermediates at these sites into vesicles…”

*Another note of caution would be to discuss the fact that unlike in the previously documented systems, one does not really have any idea what a “stimulus” means here. The ChIEF is meant to mimic the phasic type of stimuli one gets out of an action potential, but this system does not have action potentials. This is the only real weakness and prevents one from fully embracing these results. However I consider this a necessary evil at this point and I think the paper should be published in* eLife.

The reviewer is correct. Excitation in *C. elegans* motor neurons is via passive propagation and neurotransmission is likely graded (44). We added the following sentences to the Discussion: “Neurotransmission at *C. elegans* neuromuscular junctions is likely to be graded rather than driven by an action potential; however, light stimulation via channelrhodopsin has been used to study single evoked responses at these synapses (44).”

*Reviewer 2*:

We thank the reviewer for the positive review and a careful read of our manuscript.

*1) Are exocytic pits (i.e., diameter <30 nm) eliminated in UNC-13 mutants? I am assuming that the data in*
Figure 5
*refer only to large diameter pits*.

To capture exocytic intermediates, synapses must be frozen within 20 ms after stimulation. Because the timing to trigger freezing in our current setup is inconsistent from shot to shot, the 20 ms time point is very difficult to obtain. Nevertheless, synaptic vesicle fusions can be inferred from changes in the number of docked vesicles before and after fusion. We counted the number of docked vesicles in UNC-13 mutants in unstimulated samples and 50 ms after stimulation. The number of docked vesicles was slightly reduced in these animals (0.5 ± 0.1 docked vesicle/profile) compared to the non-stimulated control animals (0.78 ± 0.1 docked vesicle/profile), but the change was not statistically significant (p = 0.06). Moreover, post-synaptic currents are largely absent in UNC-13(s69) mutants, and thus we expect that there would be almost no exocytic intermediates in UNC-13 mutants. These data are consistent with the almost complete absence of evoked responses in UNC-13 mutants (Richmond, Weimer, and Jorgensen, 2001).

*2) Is there a spatial pattern to the recovery of docked SVs following stimulation? The authors very nicely show that docked SVs recover with a time constant of ∼2 seconds (*Figure 4*). It would be very interesting to know the spatial location of the newly docked SVs. Do proximal and distal docked SVs appear with distinct kinetics? You have all of these data, so it would be great to analyze the spatial pattern of docking recovery*.

To address the reviewer’s concern, we prepared figures showing the distribution of docked vesicles relative to dense projection at different time points after stimulation (Reviewers’ figure 1).

**Reviewers’ figure 1 fig8:** 

In the manuscript, we demonstrated that the replenishment of docking sites occurs between 1 s and 3 s after stimulation (Figure 4). We did not observe any spatial pattern of the recovery. However, this finding does not necessarily suggest that each docking site has the same recovery time constant. Previous studies have suggested that tethering of synaptic vesicles to the dense projection is important for proper evoked neurotransmission (Stigloher et al., 2011; Hallermann and Silver 2012), and it is tempting to speculate that the recovery of docked vesicles in the proximity of dense projection has faster kinetics. In the future, it would be interesting to analyze the locations of vesicle recovery with the finer temporal resolution between 1 s and 3 s.

*3) I did not understand how the SV docking data were plotted in the inset of Figure 2J. Does the flesh tone shading indicate the number of docked SVs at each radial position? Does the inset illustrate how the radial positions were segmented into bins? If so, the shading doesn't clearly illustrate the binning to me (or my visual acuity has declined...)*.

To illustrate how the data are binned, we made two modifications to the inset. First, we enlarged the figure so that it aligns with the binning data shown below. Second, we diagram two sample synaptic vesicles to illustrate how each would be binned.

*4) How many total exocytic pits were analyzed in Figure 2J? It would make it easier to interpret these data if the total number of pits were reported*.

The original figure (now moved to Figure 2—figure supplement 1) showed the distribution of 56 exocytic ʻomegaʼ structures from 12 fully reconstructed synapses. We added these numbers to the figure. The core of the reviewer’s concern, however, is that the distribution of fusing vesicles should be plotted so that it can be compared with the number of docked vesicles missing at 20 ms or 50 ms after stimulation. Therefore, we re-plotted the figure to show the number of vesicles instead of the percentage of fusing vesicles at the particular locations.

In addition, we only plot fusing vesicles per profile instead of per fully reconstructed synapse to match the panels in Figure 3. The black-colored line in the new graph indicates the expected number of vesicles fusing at each position as calculated from Figure 3. Specifically, these values are the difference in the number of docked vesicles between non-stimulated control and 20 ms after stimulation in each bin. Vesicle fusions are expected to be highest within 33 nm of dense projection (the first two bins), but we did not observe many fusing vesicles in this region likely because vesicles had already been collapsed into the membrane. The calcium channels are thought to be located at dense projections (Gracheva et al., 2008), and thus docked vesicles in the proximity of dense projection are likely to fuse most rapidly following a stimulation.

*5) I found it hard to discern the difference in shading between docked SVs and exocytic pits in*
Figure 2.

We changed the color scheme. Fusing vesicles are illustrated in the same color as the plasma membrane to indicate that the membrane is continuous.

*6) I found some of the p-values shown in the table in Supplementary file 1 surprising. For example, p=0.01 for total SVs in control (32) versus after stimulation (31). Is this p-value correct*?

We double-checked all p-values in the table. That was an error; we thank the reviewer for pointing it out.

*7) Is the total number of tethered SVs sufficient to account for the fast component of recovery (tau = 100 ms) in paired pulse experiments*?

The reviewer is asking how the readily releasable pool, as defined electrophysiologically, is refilled (Figure 4). There are two components to this recovery: a fast component (tau = 100 ms) and a slow component (tau = 2.2 s).

The slow component correlates with the recovery of morphologically docked vesicles. These data support previous data in *C. elegans* demonstrating that the molecularly-defined ʻprimed poolʼ of vesicles (SNARE complex), is identical to the electrophysiologically-defined ʻreadily-releasable poolʼ (evoked release), is identical to the ultrastructurally-defined ʻdocked poolʻ (membrane contact) (22). Refilling of the docked pool could come from the tethered pool of vesicles or from ʻfree vesiclesʼ in the cytoplasm (although these vesicles seem to be tethered to each other; Stigloher et al. 2011). The pool of tethered vesicles is large enough to refill the docked pool: there are 2.5 docked vesicles at rest per profile and 1.5 vesicles are lost after a stimulation. There are 3 tethered vesicles per active zone so it is possible that the tethered vesicles could refill the docked pool. Moreover, there is a statistically significant increase in the tethered pool while the docked pool is being refilled (1 second after the stimulus), and the tethered pool size is restored to its normal level when refilling of docked vesicles is complete (3 seconds). Nevertheless, the suggestion that the tethered pool refills the docked pool is still conjecture, until the tethered pool can be specifically disrupted by mutation.

The fast component of recovery of the readily releasable pool could be from four potential sources: the tethered pool, the docked pool, recovery of presynaptic currents, or recovery of postsynaptic currents.

The first possibility is that the fast recovery comes from vesicles in the tethered pool. If fast recovery were via docking, how many vesicles would this represent? From recordings, 40% of the readily releasable pool is recovered in the first 500 ms, which would correspond to 0.8 synaptic vesicles recovered per profile within this short time interval. Coincidently there is an equivalent of 0.8 tethered vesicles lost at the 50 ms and 100 ms time points (Figure 3 and Figure 4; although the reduction is not quite at the level of significance). So at first glance the data suggest the rapid phase of recovery could be contributed by docking of tethered vesicles. If that were true, we should see a fast recovery of docked vesicles as well and that is not observed (Figure 4).

Where did the tethered vesicles go? If the reduction is real, it is possible that a fraction of tethered vesicles can fuse, possibly as components of a slow releasable pool.

The second possibility is that the fast recovery comes from the docked pool. In this model, only a fraction of the docked vesicles are in the immediately releasable pool. Some vesicles might be in an incompetent state (84; 58; 32). Perhaps these remaining docked vesicles undergo a transition in the fusion machinery to become releasable, perhaps by a rearrangement of the SNARE proteins, UNC-18, complexin and synaptotagmin. We discuss this possibility extensively in the Discussion.

The third possibility is that the fast recovery comes from changes in the presynaptic ion channels. For example, calcium-activated potassium channels may limit calcium influx in subsequent stimuli and these must deactivate for recovery of the N-type channels (Dworetzky et al., 1996; Vergara et al., 1998). Alternatively, it is the calcium channels themselves that are inhibited by calcium influx (reviewed in Findeisen and Minor 2010; Su et al. 2012; Kim and Ryan 2013).

The fourth possibility is that fast recovery comes from postsynaptic receptor resensitization. Alpha acetylcholine-gated ion channels recover from desensitization of 200 ms (Mike et al., 2000), which is in the range for what we see for rapid recovery (τ = 100 ms). However, our electrophysiological experiments were done in a mutant lacking the alpha7-like ACR-16 receptor. Only the levamisole receptor was present which shows very little desensitization under high frequency stimulation (44).

*8) The data in*
Figure 3
*indicate that docked SVs <600 nm from the dense projection are lost following stimulation. Where is the adherens junction in this diagram? I found this confusing because I thought that the peri-synaptic SVs are ∼400 nm from the DP and are not supposed to undergo evoked fusion*.

On average, adherens junctions are located 250 to 350 nm from the dense projection and synaptic vesicles are not docked within the junction. In our distribution diagrams in the past the adherens junction generated a depressed region for docking – a region that we previously described as the ”vesicle free zone” (22). The position of the adherens junctions can be variable, one side of the VA or VB acetylcholine neurons that faces a muscle cell is typically longer, and in such a case, the adherens junctions may be located up to ∼1000 nm from a dense projection. Vesicles that are docked to this face undergo exocytosis, and thus synaptic vesicles docked quite distant from the dense projection can undergo fusion at those sites. In the previous submission, we simply measured the distance from the dense projection to each docked vesicle whether they were in the active zone or beyond the adherens junction in the perisynaptic zone. The variability of the position of the adherens junction caused the ʻvesicle free zoneʼ to be obscured in these figures. In this resubmission we only graphed docked vesicles in the active zone (Figure 3). Nevertheless, because some synapses are smaller than others, the number of docked vesicles declines simply because the declining number of synapses that contributed to this bin.

To clarify this point, we added the following text: “Note that the adherens junctions are located on average 250 – 350 nm from the dense projection, but in acetylcholine neurons the arrangement is asymmetric –the active zone facing the muscle can be up to ∼1000 nm. Thus, the apparent reduction in docked vesicles further than 133-660 nm is caused by the staggered ends of the active zones rather than a reduction in the density of docked vesicles. Docked vesicles in the perisynaptic zone are not counted.”

*Reviewer 3*:

We thank the reviewer for supporting our manuscript for publication in *eLife*.

*It would be important to provide further evidence that the invaginations seen poststimulation (*Figure 5*) really do represent endocytic rather than exocytic structures. Do these contain dynamin (i.e., by immunogold or by correlative SRLM/ EM)*?

The reviewer would like us to localize dynamin to the neck of the invaginations. We tried to perform this experiment using correlative nanoscopic fluorescence and electron microscopy (nano-fEM), but unfortunately this failed. There are several reasons for the failure. First, membrane contrast is too low in the absence of osmium to reliably spot invaginations. Osmium is normally used to provide contrast to membranes but cannot be used for localizing proteins because it oxidizes either the fluorescent protein in nano-fEM, or the antigen in immuno-EM and destroys the signal. Second, the membranes are blurred because nano-fEM uses a scanning electron micrograph. The resolution in SEM is limited by the diameter of the electron beam used to scan across the specimen. For our Nova Nano SEM the beam width is about 5 nm.

This means that as the beam scans across a dense membrane, the density will be blurred over 15 nm. Third, the neck of invaginations will be particularly rare in our images. Our sections are 70-80 nm thick. A scanning electron microscope only visualizes the top surface of the section, approximately 5 nm. Thus, the invaginations must be cut perfectly at the plane of the neck so that the neck is at the surface of the plastic sections. The chance of catching the neck of these structures is just too low. Thus instead of localizing dynamin to the neck of invagination, we stimulated dynamin mutants. As a reminder, endocytic structures are already trapped at the plasma membrane without stimulation in these mutants due to the endogenous activity (Figure 7). Stimulation of these neurons generated even larger vesicles (Figure 7), demonstrating that membrane from evoked fusions was trapped in invaginations at the dense projection and adherens junction.

*Whether or not invaginations near dense projections vs. adherens junctions really represent two mechanisms of endocytosis in my opinion cannot be concluded from the data. What rather appears different is their time course. How do the authors envision the function of AJs as endocytic sites? Is it conceivable that common structural determinants underlie ultrafast endocytosis at dense projections and AJs? This potential caveat should also be reflected in the title*.

We agree with the reviewer. We do not have evidence suggesting that the mechanisms for these two pathways are different. In fact, we believe that these two pathways are likely mediated by the same set of molecules. The title of our manuscript had reflected the fact that these two endocytic pathways are separated by space and time and does not suggest the difference in the molecular mechanisms. To make this point clear, we removed the reference to “two modes of endocytosis” from the title; the fact that there are two sites is clear in the Abstract.